# Prolonged T-cell activation and long COVID symptoms independently associate with severe COVID-19 at 3 months

Marianna Santopaolo[1†], Michaela Gregorova[1†], Fergus Hamilton[2], David Arnold[2], Anna Long[3], Aurora Lacey[1], Elizabeth Oliver[1], Alice Halliday[1], Holly Baum[1], Kristy Hamilton[1], Rachel Milligan[1], Olivia Pearce[3], Lea Knezevic[4], Begonia Morales Aza[1], Alice Milne[2], Emily Milodowski[4], Eben Jones[1], Rajeka Lazarus[5], Anu Goenka[1,6], Adam Finn[1,6,7], Nicholas Maskell[2], Andrew D Davidson[1], Kathleen Gillespie[3], Linda Wooldridge[4], Laura Rivino[1*]

[1]School of Cellular and Molecular Medicine, University of Bristol, Bristol, United Kingdom; [2]Academic Respiratory Unit, North Bristol NHS Trust, Bristol, United Kingdom; [3]Diabetes and Metabolism, Bristol Medical School, University of Bristol, Bristol, United Kingdom; [4]Bristol Veterinary School, University of Bristol, Bristol, United Kingdom; [5]University Hospitals Bristol and Weston NHS Foundation Trust, Bristol, United Kingdom; [6]Department of Paediatric Immunology and Infectious Diseases, Bristol Royal Hospital for Children, Bristol, United Kingdom; [7]School of Population Health Sciences, University of Bristol, Bristol, United Kingdom

*For correspondence:
laura.rivino@bristol.ac.uk

†These authors contributed equally to this work

Competing interest: The authors declare that no competing interests exist.

**Abstract** Coronavirus disease-19 (COVID-19) causes immune perturbations which may persist long term, and patients frequently report ongoing symptoms for months after recovery. We assessed immune activation at 3–12 months post hospital admission in 187 samples from 63 patients with mild, moderate, or severe disease and investigated whether it associates with long COVID. At 3 months, patients with severe disease displayed persistent activation of CD4+ and CD8+ T-cells, based on expression of HLA-DR, CD38, Ki67, and granzyme B, and elevated plasma levels of interleukin-4 (IL-4), IL-7, IL-17, and tumor necrosis factor-alpha (TNF-α) compared to mild and/or moderate patients. Plasma from severe patients at 3 months caused T-cells from healthy donors to upregulate IL-15Rα, suggesting that plasma factors in severe patients may increase T-cell responsiveness to IL-15-driven bystander activation. Patients with severe disease reported a higher number of long COVID symptoms which did not however correlate with cellular immune activation/pro-inflammatory cytokines after adjusting for age, sex, and disease severity. Our data suggests that long COVID and persistent immune activation may correlate independently with severe disease.

## Editor's evaluation

This is an important paper that presents convincing evidence that certain cellular and molecular immune fingerprints at day 30 post-infection are associated with severe prior infection and may be risk factors for prolonged symptoms. The work will be of broad interest to clinicians, immunologists, and virologists.

## Introduction

Infection with severe acute respiratory syndrome *coronavirus 2* (SARS-CoV-2), the causative agent of coronavirus disease-19 (COVID-19), can be asymptomatic or lead to a broad spectrum of disease manifestations from mild to severe disease and death. There is evidence showing that acute COVID-19 causes a profound activation of innate and adaptive immune cells, and distinct immunological signatures are shown to correlate with disease outcomes (*Mathew et al., 2020*). Viral control and mild disease associate with the rapid generation of T-cells and antibodies targeting SARS-CoV-2 (*Lucas et al., 2021*; *Tan et al., 2021*), while severe disease is characterised by immune cell hyperactivation and a 'cytokine storm' (*Chen et al., 2020*; *Kuri-Cervantes et al., 2020*; *Ruan et al., 2020*). Commonly observed biomarkers of severe COVID-19 include lymphopenia, increased neutrophil to T-cell ratio, and elevated levels of pro-inflammatory cytokines/mediators such as interleukin-6 (IL-6), IL-10, IL-17, monocyte chemoattractant protein-1 (MCP-1), interferon gamma-induced protein 10 (IP-10), CRP, IL-1Rα, and IL-1β. In a smaller proportion of individuals, severe COVID-19 may be driven by pre-existing anti-interferon gamma (anti-IFN-γ) autoantibodies (*Bastard et al., 2020*; *Liu et al., 2020*; *Lucas et al., 2020*; *Zhao et al., 2020*). Other distinctive features of severe disease include a dysregulation in the myeloid compartment consisting of increased circulating immature/dysfunctional neutrophils and immature Ki67$^+$ COX-2$^{low}$ monocytes (*Mann et al., 2020*; *Schulte-Schrepping et al., 2020*; *Wu et al., 2020*; *Mardi et al., 2021*), while T-cell activation and proliferation appears to be heterogenous in severe patients (*Mathew et al., 2020*; *Kuri-Cervantes et al., 2020*).

The kinetics of immune recovery following the changes occurring during acute COVID-19 are complex and not fully understood. Longitudinal studies show that immune abnormalities and inflammation may persist after severe COVID-19, with highly activated myeloid cells, pro-inflammatory cytokines, and persistently activated T-cells detected 8–12 months after COVID-19 (*Bergamaschi et al., 2021*; *Phetsouphanh et al., 2022*; *Taeschler et al., 2022*).

The potential impact of prolonged immune activation post recovery and whether it may underlie the symptoms of post-acute sequelae of SARS-CoV-2 infection or long COVID - a syndrome characterised by long-lasting symptoms affecting multiple organs - remains unclear. Long COVID is observed in 8–21% of individuals following mild to severe COVID-19, although a higher prevalence of symptoms is reported in patients requiring ICU admission and/or mechanical ventilation (*Alwan, 2021*; *Nalbandian et al., 2021*; *Ballering et al., 2022*). Several studies have reported a correlation between long COVID symptoms and a variety of immune profiles including a more rapid decline of SARS-CoV-2 memory CD8$^+$ T-cells and lower levels of cytotoxic SARS-CoV-2 N-specific CD8$^+$ T-cells (*Peluso et al., 2021*), increased levels of cytotoxic CD8$^+$ T-cells, and elevated plasma levels of type I cytokines, IL-1β and IL-6 (*Shuwa et al., 2021*; *Schultheiß et al., 2022*). Another longitudinal study conducted in 69 patients found long COVID to associate with an altered immune blood cell transcriptome which included significant changes in genes involved in cell cycle, but not with the phenotypic profiles of these cells (*Ryan et al., 2022*). A large UK cohort study (PHOSP-COVID) of hospitalised COVID-19 patients identified female sex, obesity, and invasive mechanical ventilation as factors associated with a lower likelihood of full recovery, with an overall recovery rate of 28.9% at both 5 and 12 months across all disease severities (*Evans et al., 2022*). Similarly, other cohort studies have identified associations between poor recovery from COVID-19 and demographic factors such as sex and age, but also highlight a possible role of immune dysregulation that occurs in older age, obesity and asthma, which warrants further investigation (*Sneller et al., 2022*; *Halpin et al., 2021*; *Thompson et al., 2022*). Other hypotheses proposed to explain the mechanisms underlying long COVID include unresolved lung tissue damage, the persistence of SARS-CoV-2 infection, the reactivation of latent viruses such as human cytomegalovirus (CMV) and Epstein-Barr virus and autoimmunity (*Proal and VanElzakker, 2021*; *Gatto et al., 2022*; *Plüß et al., 2021*; *Zollner et al., 2022*). Due to the limited effectiveness of COVID-19 vaccines in protecting from re-infection, SARS-CoV-2 continues to circulate within our populations and millions of people worldwide are at risk of experiencing long-term health complications from COVID-19. Studies suggest that COVID-19 vaccination may reduce the risk of developing long COVID (*Al-Aly et al., 2022*; *Notarte et al., 2022*). Nevertheless the burden of long COVID threatens to increase and further compromise our post-pandemic economic recovery. A better understanding of the mechanisms underlying long COVID and the design/repurposing of drugs to treat or prevent this syndrome is urgently needed.

In this study we investigated whether patients who had COVID-19, 3 months post hospitalisation display persistent immune activation and ongoing inflammation and if there are potential links between the observed immune profiles, COVID-19 disease severity and long COVID symptoms. We also investigated whether persistent immune activation and disease severity have an impact on the capacity of patients to generate and maintain a memory T-cell response and antibodies to SARS-CoV-2. Our immunological analysis of 187 samples from 63 hospitalised patients recovering from mild, moderate, or severe disease showed increased levels of activated CD4[+] and CD8[+] T-cells and increased plasma levels of T-cell-related cytokines (IL-4, IL-7, IL-17, and tumor necrosis factor-alpha [TNF-α]) at 3 months in severe compared to moderate/mild patients. T-cell activation and cytokine levels decreased at 12 months and were comparable in mild, moderate, and severe patients. The SARS-CoV-2-specific memory T-cell response to spike, membrane and nucleocapsid was robust and qualitatively similar across the disease severities, however the magnitude of the CD4[+] and CD8[+] T-cell and antibody response was higher in patients with moderate compared to mild disease, as were the plasma levels of IFN-γ, a key cytokine for anti-viral immunity. Long COVID symptoms at 3 months were reported in 80% of patients across all disease severities but were more frequent in patients with severe disease. Poisson regression analysis showed a significant association between ongoing symptoms and the frequency of a subset of moderately activated CD4[+] T and non-cytotoxic HLA-DR[+] CD8[+] T-cells in unadjusted analyses, but these associations did not meet the criteria for statistical significance after adjusting for age, sex, and severity grades. Our results demonstrate the presence of immune perturbations in peripheral blood CD4[+] and CD8[+] T-cell populations in severe COVID-19 patients 3 months after hospitalisation with no direct association between long COVID symptoms and immune activation/pro-inflammatory cytokines, for the markers that were measured.

## Results
### DISCOVER patients

To investigate immune profiles in convalescent COVID-19 patients, we obtained peripheral blood mononuclear cells (PBMCs) and plasma from 63 patients enrolled in the DISCOVER study at 3 months post admission for COVID-19, and where possible matched PBMCs at 12 months and plasma samples at 8 and 12 months. The demographics and clinical characteristics of the 63 patients included in this study are shown in *Table 1*. Patients were stratified into mild, moderate, and severe based on the type of respiratory support they required during the acute illness, as follows: mild patients did not require supplementary oxygen or intensive care; moderate patients required supplementary oxygen during admission, and severe patients required invasive mechanical ventilation, non-invasive ventilation, and/or admission to the intensive treatment unit. Of the 63 patients included in this analysis 17, 32, and 14 recovered from mild, moderate, and severe disease, respectively. Overall, a higher proportion of patients were male (40/63) and the median age (± SD) of patients was 53±14.5, 58±12.6, and 61.5±10 years for mild, moderate, and severe patients, respectively. Patient ethnicity was predominantly Caucasian with an Asian/Black minority. Body mass index (BMI) was largely within the unhealthy range across all disease severities (overweight, obese, and extremely obese) with 7.6%, 15.6%, and 14.3% of patients with respectively mild, moderate, and severe disease displaying healthy BMI. The comorbidities observed in these 63 patients were consistent with those known to be associated with higher risks for COVID-19 hospitalisation and included heart disease, diabetes (predominantly type-2), hypertension, and chronic lung disease. Overall, the percentage of patients with comorbidities tended to increase progressively with disease severity (mild: 52.9%; moderate: 56.2%, and severe: 85.7%). The duration of hospital admission ranged from 3.3±1.99 to 7.8±5.16 and 12±6.67 days (± SD) in mild, moderate, and severe patients, respectively. None of the patients had received COVID-19 vaccination at the time of blood collection.

### Increased CD4[+] and CD8[+] T-cell activation in severe patients at 3 months

To investigate the presence of ongoing immune activation following recovery from COVID-19, we assessed the phenotype and activation/proliferation profiles of conventional CD4[+] and CD8[+] T-cells, TCR-γδ T, NK, B-cells, and monocytes by flow cytometry in PBMC samples from 63 patients at 3 months post admission. T and NK cells were assessed for expression of markers of activation, differentiation,

**Table 1.** Details of the patients included in the immunological analysis of this study.

| Disease severity (n) | | Mild | Moderate | Severe |
|---|---|---|---|---|
| | | (n=17) | (n=32) | (n=14) |
| Age (median, SD) | | 53±14.5 | 58±12.6 | 61.5±10 |
| Sex, % (n) | Female | 35.3 (6) | 31.2 (10) | 50 (7) |
| | Male | 64.7 (11) | 68.5 (22) | 50 (7) |
| Ethnicity, % (n) | Caucasian | 83.3% (14) | 78.1% (25) | 87.7% (12) |
| | Asian | 11.7% (2) | 12.5% (4) | 14.3% (2) |
| | Black | 5.9% (1) | 6.3% (2) | 0 (0) |
| | Missing data | 0 (0) | 3.1% (1) | 0 (0) |
| BMI, kg/m², % (average, SD) | Healthy | 7.6% (22.3±0.57) | 12.5% (23.75±0.5) | 14.3% (23±0) |
| | Overweight | 35.3% (26.6±1.5) | 28.1% (27±1.39) | 28.6% (28.75±0.5) |
| | Obese | 35.3% (32.8±1.47) | 46.9% (32.7±2.46) | 28.6% (35±2.1) |
| | Extremely obese | 11.8% (61±28.2) | 9.38% (46.3±6.02) | 28.6% (46±7.34) |
| | Missing data | 0 | 3.1% | 0 |
| Comorbidity % (n) | None | 47.1% (8) | 43.75% (14) | 14.3% (2) |
| | Heart disease | 23.5% (4) | 15.6% (5) | 14.3% (2) |
| | T1DM | 0 (0) | 3.1% (1) | 7.14% (1) |
| | T2DM | 5.88% (1) | 12.5% (4) | 14.3% (2) |
| | Hypertension | 17.65 (3) | 18.75% (6) | 50% (7) |
| | Chronic lung disease | 5.88% (1) | 21.88% (7) | 50% (7) |
| | Kidney disease | 5.88 (1) | 9.37% (3) | 14.3% (2) |
| | Mental health | 0 (0) | 6.25% (2) | 14.3% (2) |
| | Cancer | 5.88% (1) | 3.1% (1) | 7.14 (1) |
| | Asthma | 5.88% (1) | 0 (0) | 14.3% (2) |
| | Total obesity | 47.1% (8) | 53.1% (17) | 57.1% (8) |
| | Other | 47.1% (8) | 28.1% (9) | 50% (7) |
| Hospital stay (average days, SD) | | 3.3±1.99 | 7.8±5.16 | 12±6.67 |
| Ongoing symptoms at 3 months (n, %) | None | 3 (17.65%) | 8 (25%) | 2 (14.3%) |
| | Dyspnoea | 7 (41.2%) | 15 (47%) | 9 (64.3%) |
| | Excessive fatigue | 5 (29.4%) | 12 (37.5%) | 9 (64.3%) |
| | Muscle weakness | 3 (17.65) | 7 (22%) | 5 (35.7%) |
| | Sleeping difficulties | 3 (17.65) | 6 (18.75%) | 8 (57%) |
| | Psychiatric | 2 (11.8%) | 8 (25%) | 6 (42.9%) |
| | Anosmia | 2 (11.8%) | 4 (12.5%) | 3 (21.4%) |
| | Chest pain | 2 (11.8%) | 6 (18.75%) | 2 (14.3%) |
| | Cough | 2 (11.8%) | 4 (12.5%) | 1 (7.14%) |
| | Other | 5 (29.4%) | 4 (12.5%) | 3 (21.4%) |

proliferation, and cytotoxicity (HLA-DR, CD38, CD69, CCR7, CD45RA, CXCR3, Ki67, granzyme B, CD56) after gating on single live cells, $CD3^+$ $CD4^+$ or $CD3^+$ $CD8^+$ T-cells and $CD3^-$ $CD56^{bright}$ $CD16^{+/-}$ or $CD56^{dim}$ $CD16^+$ NK cell populations, respectively (*Figure 1*, gating strategies in *Figure 1—figure supplement 1*; list of antibodies in Key resources table). Data were analysed by FlowJo using a manual gating strategy as well as the dimensionality reduction algorithm uniform manifold approximation and projection (UMAP) for dimension reduction and cluster analysis using FlowSOM.

Overall, at 3 months post admission patients' blood lymphocyte and neutrophil counts, CRP, and albumin levels - which were mostly perturbated during the acute illness - had normalised to levels that remained similar at 8 months, suggesting a resolution of the peak inflammatory events occurring during the acute illness (*Figure 1—figure supplement 2*). The frequencies and absolute numbers of $CD4^+$ T-cells were similar in patients across disease severities (*Figure 1A and B*). However, we observed an increased frequency of $CD4^+$ T-cells expressing the peripheral homing receptor CXCR3 in severe compared to moderate patients and an increased frequency of $CD4^+$ T-cells co-expressing the proliferation/activation markers Ki67 and CD38 in severe compared to mild patients (*Figure 1C and D*). The overall differentiation status of $CD4^+$ T-cells, defined by the coordinated expression of CCR7 and CD45RA, was similar in mild, moderate, and severe patients, with high proportions of naïve ($CCR7^+$ $CD45RA^+$) and T central memory cells ($CCR7^+$ $CD45RA^-$, $T_{CM}$) followed by detectable levels of T effector memory cells ($CCR7^-$ $CD45RA^-$, $T_{EM}$) and low frequencies of T effector memory re-expressing RA cells ($CCR7^-$ $CD45RA^+$, $T_{EMRA}$) (*Figure 1E*). The most abundant naïve and $T_{CM}$ $CD4^+$ T-cell subsets contained significantly higher levels of cells co-expressing markers of activation, proliferation, and cytotoxic potential ($HLA-DR^+$ $CD38^+$, $HLA-DR^+$ $Ki67^+$, $CD38^+$ granzyme $B^+$) in severe compared to moderate/mild patients (*Figure 1F–K*). An unsupervised analysis by UMAP and FlowSOM revealed the presence of 15 distinct clusters of cells characterised by a unique, coordinated expression of the analysed markers (*Figure 1L*), shown as mean fluorescence intensity in the heatmap (*Figure 1M*). These analyses revealed significantly higher frequencies in severe compared to moderate patients of clusters of cells expressing HLA-DR, CD38, and Ki67 (populations 8 and 13) as well as of cells that express CCR7 but lack expression of all other markers analysed, and which resemble undifferentiated, resting $CD4^+$ T-cells (population 2) (*Figure 1N*). The latter population accounted for a large proportion of $CD4^+$ T-cells (up to 60% of cells in severe patients). A population characterised by low levels of expression of all markers analysed is decreased in severe compared to moderate patients (population 1, *Figure 1N*). In summary, both manual gating and unsupervised analysis reveal the presence of higher frequencies of $CD4^+$ T-cell populations expressing markers of activation, proliferation, and peripheral tissue homing in patients who recovered from severe versus mild and/or moderate disease.

Similarly, the analyses of $CD8^+$ T-cells showed comparable frequencies and absolute numbers of circulating $CD8^+$ T-cells in patients across disease severities and increased levels of $CD8^+$ T-cells expressing markers of activation ($HLADR^+CD38^+$) in severe compared to moderate patients and cytotoxicity (granzyme $B^+$) in severe compared to mild patients (*Figure 2A–D*). The overall differentiation status of $CD8^+$ T-cells was comparable in patients across disease severities with detectable frequencies of naïve, $T_{CM}$, $T_{EM}$, and $T_{EMRA}$ cells (*Figure 2E*). $CD8^+$ T-cells expressing markers of activation and cytotoxicity ($HLADR^+$ $CD38^+$ and $CD38^+$ granzyme $B^+$) could be detected at higher frequencies within $T_{CM}$, $T_{EM}$, and $T_{EMRA}$ subsets in severe compared to moderate and/or mild patients (*Figure 2F–I*). Unsupervised analysis by UMAP and FlowSOM of $CD8^+$ T-cells showed the presence of 15 clusters of cells expressing the analysed markers (*Figure 2J and K*). The frequencies of a cluster of $CD8^+$ T-cells expressing high levels of granzyme B and CD45RA (cluster 3) were significantly increased in both severe and moderate compared to mild patients and represented up to 50% of all $CD8^+$ T-cells in severe/moderate groups and up to 30% in mild group (*Figure 2L*). $CD8^+$ T-cells expressing high levels of CCR7 and intermediate levels of granzyme B, CXCR3, and CD56 were increased significantly in both mild and severe compared to moderate patients (cluster 12). In summary, manual and unsupervised analysis revealed the presence of $CD8^+$ T-cells expressing markers of activation and cytotoxicity which are increased in severe compared to moderate and/or mild patients. Analysis of the T-cell phenotypes in the same patients at 12 months showed that $CD4^+$ and $CD8^+$ T-cell activation and proliferation decreased significantly from 3 to 12 months and activated/proliferating T-cells were largely absent at 12 months (*Figure 2—figure supplement 1*). Analysis by manual gating showed a significant decrease of activated/proliferating $CD4^+$ and $CD8^+$ T-cells ($Ki67^+$ $CD38^+$, $HLA-DR^+$ $CD38^+$) and of $CD4^+$ and $CD8^+$ T-cells expressing the peripheral tissue-homing receptor $CXCR3^+$ from 3 to 12 months

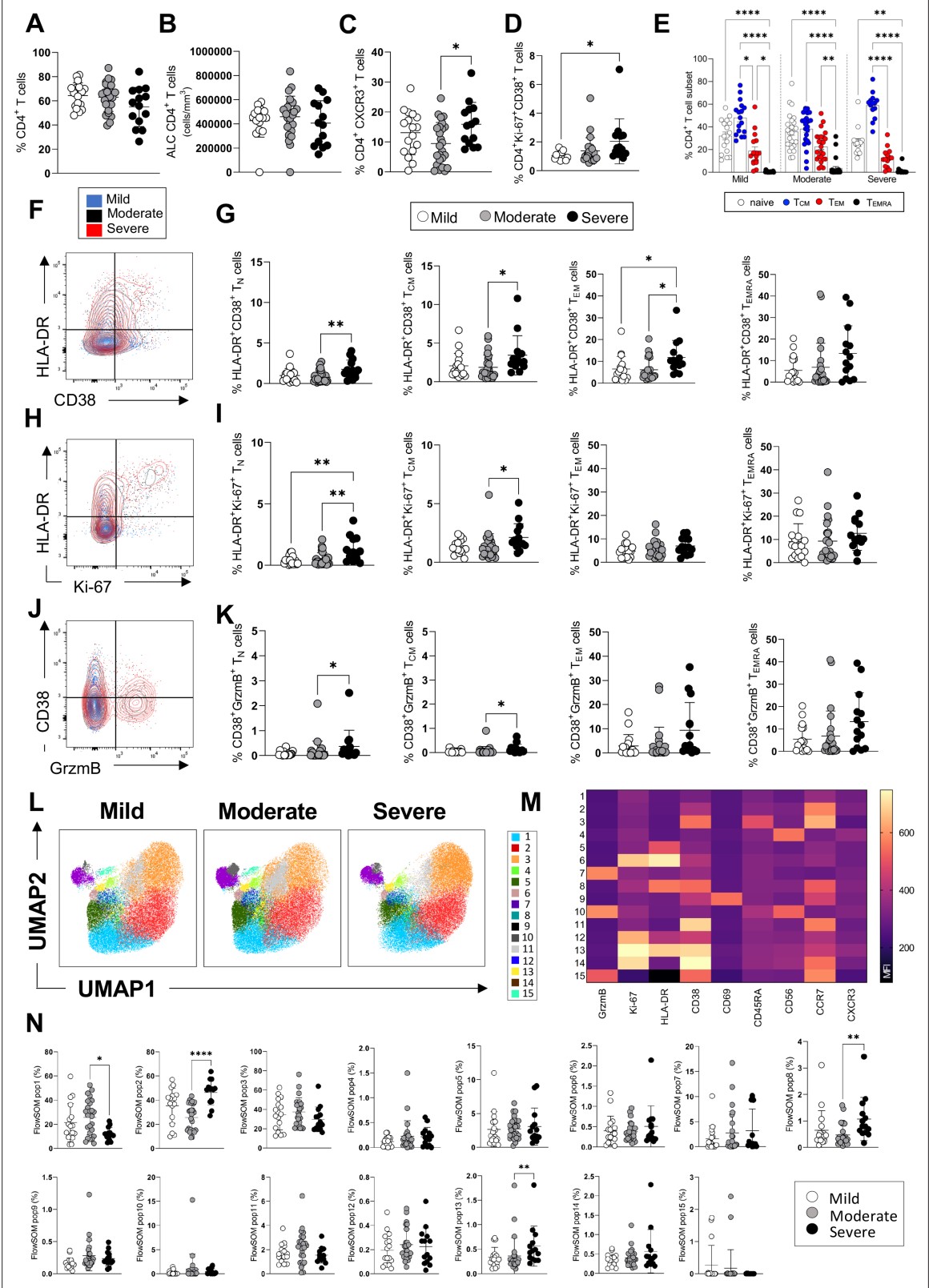

**Figure 1.** CD4+ T-cell profiles in convalescent coronavirus disease-19 (COVID-19) patients at 3 months post admission. (**A–D**) Percentage of CD4+ T-cells within the CD3+ gate (**A**), absolute number of CD4+ T-cells (cells/mm$^3$) (**B**), and percentages of CD4+ T-cells expressing CXCR3 (**C**) and co-expressing Ki67/CD38 (**D**) are shown in mild, moderate, and severe patients. (**E**) Percentages of naïve (CCR7+ CD45RA+), T central memory (T$_{CM}$, CCR7+ CD45RA-), T effector memory (T$_{EM}$, CCR7- CD45RA-), and T effector memory RA re-expressing (T$_{EMRA}$, CCR7- CD45RA+) CD4+ T-cells are shown for patients with mild,

*Figure 1 continued on next page*

*Figure 1 continued*

moderate, and severe disease. (**F**) Flow cytometry plot showing a representative staining from a mild, moderate, and severe patient of HLA-DR and CD38 expression in CD4$^+$ T$_{EM}$ cells (overlaid and shown respectively in blue, black, and red). (**G**) Percentages of activated HLA-DR$^+$CD38$^+$ CD4$^+$ T-cells within naïve, T$_{CM}$, T$_{EM}$, and T$_{EMRA}$ cells. (**H**) Flow cytometry plot with a representative staining from a mild, moderate, and severe patient of HLA-DR and Ki67 expression in CD4$^+$ T$_{EM}$ cells. (**I**) Percentages of proliferating HLA-DR$^+$ Ki67$^+$ CD4$^+$ T-cells within naïve, T$_{CM}$, T$_{EM}$, and T$_{EMRA}$ cells. (**J**) Flow cytometry plot with a representative staining from a mild, moderate, and severe patient of HLA-DR and granzyme B (GrzmB) expression in CD4$^+$ T$_{EM}$ cells. (**K**) Percentages of proliferating HLA-DR$^+$ GrzmB$^+$ CD4$^+$ T-cells within naïve, T$_{CM}$, T$_{EM}$, and T$_{EMRA}$ cells. (**L**) Unsupervised uniform manifold approximation and projection (UMAP) analysis showing the FlowSOM clusters in mild (N=17), moderate (N=25), and severe (N=14) patients. Plots are gated on CD4$^+$ T-cells. (**M**) Heatmap with the expression of each analysed marker within the FlowSOM populations shown as mean fluorescence intensity (MFI). (**N**) Summary of the percentage of CD4$^+$ T-cells within the indicated FlowSOM populations in mild, moderate, and severe patients. Data in graphs are visualised as mean ± SEM. Statistics are calculated by one-way ANOVA (Kruskal-Wallis test) with Dunn's correction for multiple testing.

The online version of this article includes the following figure supplement(s) for figure 1:

**Figure supplement 1.** Gating strategy used to identify CD4$^+$, CD8$^+$, and TCR-γδ T-cells, NK cells, and monocytes.

**Figure supplement 2.** Dynamic changes of immune populations and inflammatory markers in coronavirus disease-19 (COVID-19) patients at acute illness, 3 and 8 months post admission.

---

(*Figure 2—figure supplement 1B-C,G-J* ). An unsupervised analysis by UMAP revealed similar distribution of T-cell clusters in patients with mild, moderate, and severe disease at 12 months, while major differences were observed between patients from the three groups at 3 months (*Figure 2—figure supplement 1K*). Our data suggests that the ongoing T-cell activation observed at 3 months in patients with severe disease had recovered by 12 months post admission to levels similar to those observed in patients with mild and moderate disease.

Analysis of expression of the same markers of activation/proliferation in NK CD56$^{dim}$ and CD56$^{bright}$ cells did not show evidence of ongoing NK cell activation and NK cells were similar in patients who recovered from mild, moderate, and severe disease (*Figure 2—figure supplement 2A-F*). Similarly, analysis of the frequencies and expression of activation markers (CD80, CD68) of classical (CD14$^+$ CD16$^-$), intermediate (CD14$^+$CD16$^+$), and non-classical (CD14$^-$CD16$^+$) monocytes showed similar frequencies of these cells across the disease severities and lack of ongoing activation (*Figure 2— figure supplement 2G-K*). Similar frequencies of unconventional TCR-γδ T-cells were detected in patients with mild, moderate, and severe disease and these cells lacked expression of markers of activation/proliferation (*Figure 2—figure supplement 2L-N*). In contrast, CD19$^+$ B cells from severe patients showed an increase in the expression of the activation marker CD80 compared to patients with mild disease, although expression of other markers of activation/proliferation did not differ (HLA-DR, CD38, and Ki67) (*Figure 2—figure supplement 2O-R*).

In summary, both manual and unsupervised analysis showed increased frequencies of activated/ proliferating CD4$^+$ and CD8$^+$ T-cells in severe compared to mild and/or moderate patients at 3 months post admission, suggesting the presence of ongoing immune activation in these patients. We did not find significant activation of other immune cells analysed (NK, TCR-γδ, monocytes), with the exception of CD19$^+$ B cells from severe patients which expressed increased levels of CD80 compared to mild patients.

## Elevated pro-inflammatory cytokines/chemokines in severe patients at 3 months

To gain insights into the factors that may be driving CD4$^+$ and CD8$^+$ T-cell activation and proliferation and define the nature of any persistent inflammation in these patients, we investigated the presence of soluble circulating pro-inflammatory cytokines/chemokines in the plasma of COVID-19 patients 3 months after admission. The plasma levels of the following 23 cytokines/chemokines were measured using a Luminex platform: granulocyte macrophage colony stimulating factor (GM-CSF), IFN-γ, IFN-α, IL-1α, IL-1β, IL-10, IL-12 p70, IL-13, IL-15, IL-17A, IL-18, IL-2, IL-4, IL-5, IL-6, IL-7, IL-8, IP-10, MCP-1, macrophage inflammatory protein-1α (MIP-1α), MIP-1β, and TNF-α (Human Procarta plex). Our results show that at 3 months the levels of IL-4, IL-7, IL-17, and TNF-α were significantly increased in the plasma of patients with severe compared to mild and/or moderate disease, while IFN-γ was increased in moderate compared to mild patients. Interestingly, IFN-γ was largely undetectable in the plasma of severe patients (*Figure 3A*). A trend for increased levels of IL-18 was observed in the plasma of severe compared to mild and moderate patients but the difference was not significantly different. The plasma

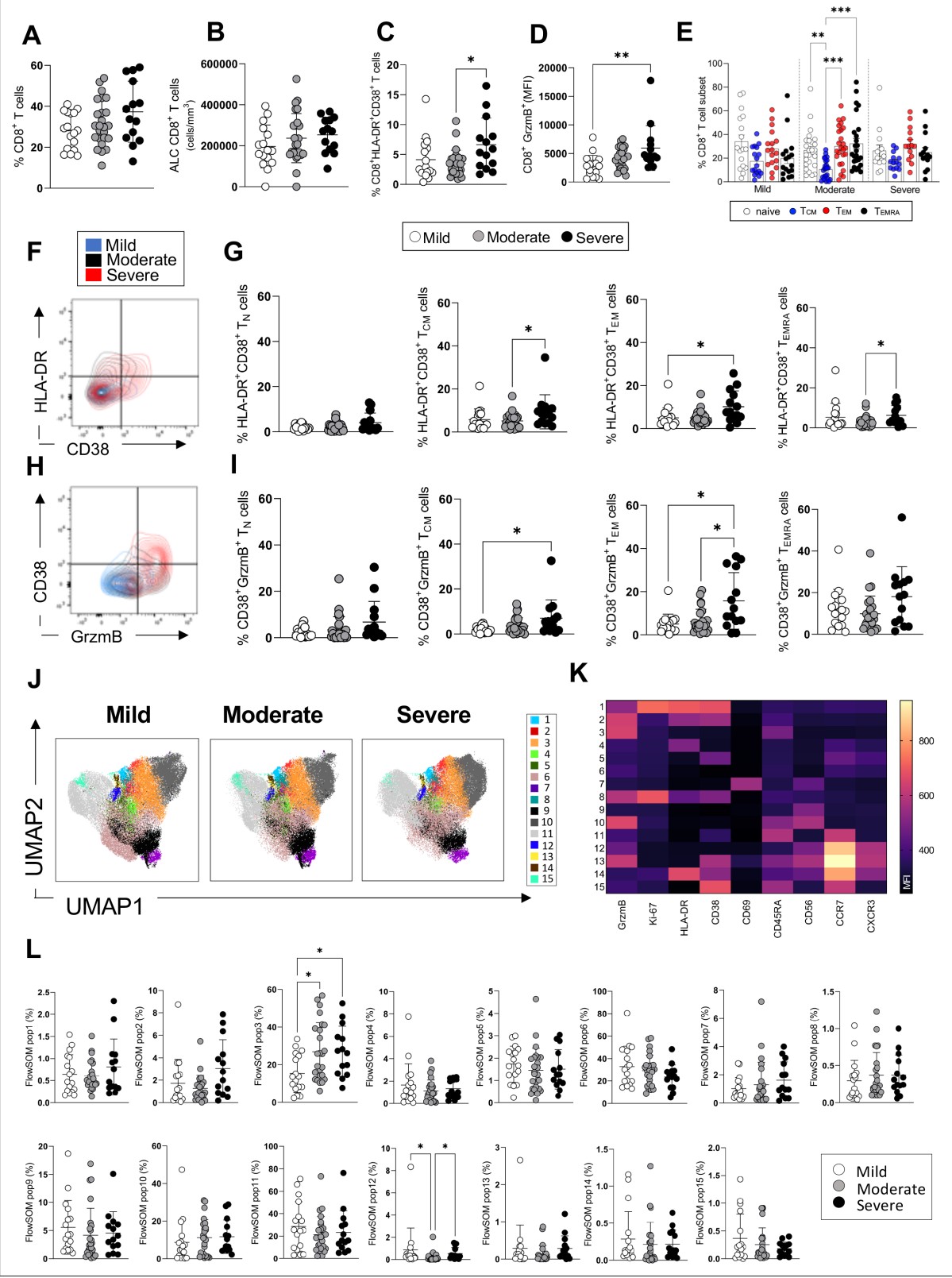

**Figure 2.** CD8⁺ T-cell profiles in convalescent coronavirus disease-19 (COVID-19) patients at 3 months post admission. (**A–D**) Percentage of CD8⁺ T-cells within the CD3⁺ gate (**A**), absolute number of CD8⁺ T-cells (cells/mm³) (**B**), and percentages of CD8⁺ T-cells co-expressing the activation markers HLA-DR/CD38 (**C**) or granzyme B (D, shown as mean fluorescence intensity [MFI]) are shown in mild, moderate, and severe patients. (**E**) Percentages of naïve (CCR7⁺ CD45RA⁺), T central memory (T_CM, CCR7⁺ CD45RA⁻), T effector memory (T_EM, CCR7⁻ CD45RA⁻), and T effector memory RA re-expressing

*Figure 2 continued on next page*

*Figure 2 continued*

($T_{EMRA}$, CCR7- CD45RA+) CD8+ T-cells in patients with mild, moderate, and severe disease. (**F**) Flow cytometry plot with a representative staining from a mild, moderate, and severe patient (overlaid and shown respectively in blue, black, and red) of HLA-DR and CD38 expression in CD8+ $T_{EM}$ cells. (**G**) Percentages of activated HLA-DR+ CD38+ CD8+ T-cells within naïve, $T_{CM}$, $T_{EM}$, and $T_{EMRA}$ cells. (**H**) Flow cytometry plot with a representative staining from a mild, moderate, and severe patient of HLA-DR and granzyme B (GrzmB) expression in CD8+ $T_{EM}$ cells. (**I**) Percentages of proliferating HLA-DR+ GrzmB+ CD8+ T-cells within naïve, $T_{CM}$, $T_{EM}$, and $T_{EMRA}$ cells. (**J**) Unsupervised uniform manifold approximation and projection (UMAP) analysis showing the FlowSOM clusters in mild (N=17), moderate (N=25), and severe (N=14) patients. Plots are gated on CD8+ T-cells. (**K**) Heatmap with MFI levels for each analysed marker within the FlowSOM populations. (**L**) Summary of percentage of CD8+ T-cells within the indicated FlowSOM populations in mild, moderate, and severe patients. Data in the graphs are shown as mean ± SEM. Statistics were calculated by one-way ANOVA (Kruskal-Wallis test) with Dunn's correction for multiple testing.

The online version of this article includes the following figure supplement(s) for figure 2:

**Figure supplement 1.** Resolution of T-cell activation at 12 months.

**Figure supplement 2.** Immune cell populations in coronavirus disease-19 (COVID-19) patients at 3 months post admission.

levels of other cytokines/chemokines tested did not differ between patients with different disease severities at 3 months (*Figure 3—figure supplement 1*). We next investigated the kinetics of expression of these cytokines in patients at 3, 8, and 12 months post admission and observed a statistically significant decrease of plasma levels of IL-4, IL-12, IL-13, and TNF-α from 3 to 8 and/or 12 months, suggesting that at 3 months the levels of these cytokines may still be affected by events occurring during the acute phase of infection. IL-15 levels were similar at 3 and 8 months and decreased thereafter (*Figure 3B and C*). IL-17 levels were higher at 3 months and progressively diminished over time, although differences were not statistically significant. In contrast, IP-10 levels were higher at 8 and 12 months compared to 3 months. The plasma levels of the other cytokines analysed including IL-7, IFN-γ, IL-18, and MCP-1 were comparable at 3–12 months post admission.

Based on our results in *Figures 1–2* showing increased levels of circulating activated/proliferating CD4+ and CD8+ T-cells in severe patients and on the elevated levels of cytokines involved in T-cell proliferation (IL-7, IL-15, TNF-α) at 3 months, we asked whether factors present in the plasma of severe patients might render T-cells more susceptible to 'bystander T-cell' activation, which occurs during viral infection and is believed to be driven by IL-15. To address this, we co-cultured purified T-cells from healthy donors (n=4) for 7 days with plasma derived from either heterologous healthy donors (n=4, 'HD plasma') or patients with mild or severe COVID-19 at 3 and 12 months (n=4, 'mild plasma'; n=4, 'severe plasma'). We observed a significant upregulation of the IL-15Rα-chain, the IL-15R subunit that binds directly to IL-15, in T-cells co-cultured in the presence of plasma from severe patients at 3 months compared to healthy donors (*Figure 3D and E*). In contrast, serum from severe patients at 12 months failed to upregulate IL-15Rα-chain. These experiments suggest that 3 months after severe COVID-19, peripheral blood CD4+ and CD8+ T-cells are exposed in vivo to plasma factors that make them more responsive to IL-15 by inducing upregulation of the IL-15R, and that persistent T-cell activation observed ex vivo at 3 months may be driven by cytokines such as IL-15.

In summary, at 3 months post admission we observe increased levels of IL-4, IL-7, IL-17, and TNF-α in the plasma of severe compared to mild and/or moderate patients. The levels of IL-4, TNF-α, and IL-17 gradually diminished from 3 to 12 months suggesting a resolution in the perturbations of immune cells that are secreting these cytokines, while those of IL-7 were comparable across these time points. Consistently, CD4+ and CD8+ T-cell activation also decreased from 3 to 12 months to levels that were largely undetectable at 12 months (*Figure 2—figure supplement 1*).

## Robust SARS-CoV-2 memory T-cell and antibody responses

To investigate the magnitude and functional features of SARS-CoV-2-specific T-cells and address potential associations with disease severity and long COVID, we performed IFN-γ Enzyme-Linked Immune absorbent Spot (ELISpot) assays with PBMC samples from 61 patients. IFN-γ production by ELISpot was measured in PBMCs after an overnight stimulation with overlapping 15-mer peptide pools spanning the sequences of the SARS-CoV-2 structural proteins spike (divided into two pools: spikes 1 and 2, which include peptides spanning the N and C terminal regions of spike, respectively), membrane and nucleocapsid. To investigate in parallel the T-cell response to a persistent virus that is commonly present in the UK population, PBMCs were stimulated with a 15-mer peptide pool spanning the human CMV pp65 protein. Positive (PMA/ionomycin) and negative control wells were included in the

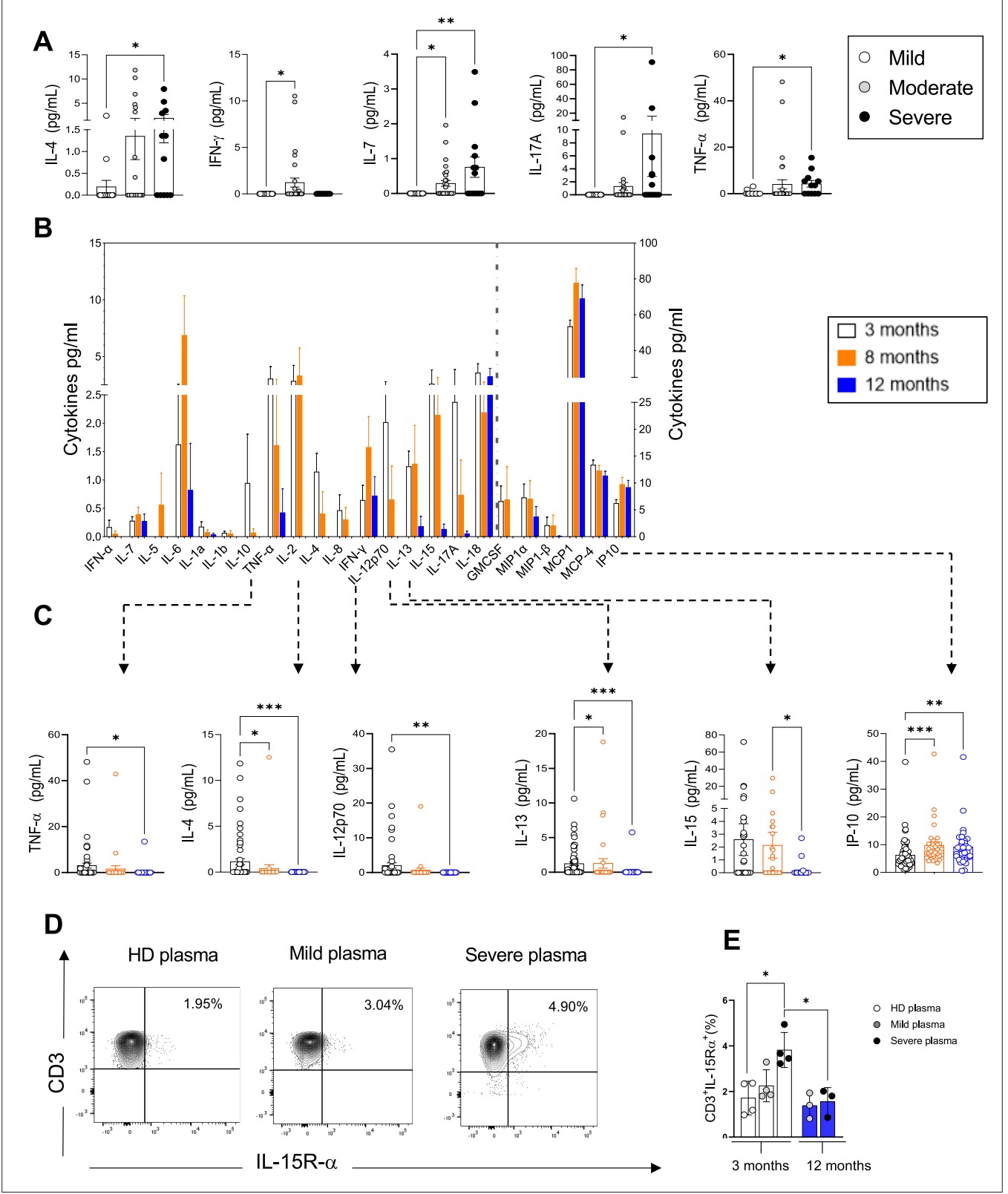

**Figure 3.** Plasma pro-inflammatory cytokines/chemokines measured at 3, 8, and 12 months. (A) Plasma cytokines/chemokines measured at 3 months post admission which differed significantly between patients with mild, moderate, and severe disease are shown (N=63: mild: N=17; moderate: N=32; severe: N=14, depicted in white, grey, and black bars, respectively). (B–C) Cytokines/chemokines measured longitudinally in matched samples in patients at 3 (n=63), 8, and 12 months post admission (n=33 samples for each time point) are shown. Data from analytes that differed significantly

*Figure 3 continued on next page*

*Figure 3 continued*

between time points in B are shown in C for each patient. (**D, E**) Purified CD3+ T-cells from healthy donors (N=4 for 3 months; N=3 for 12 months) were co-cultured with plasma from 4 healthy donors, 4 mild, and 4 severe patients *at 3 months post infection.* Shown is IL-15R-α expression in T-cells from a representative donor at 3 months (**D**) and the average expression of IL-15Rα by T-cells from each peripheral blood mononuclear cell (PBMC) donor after co-culture with plasma from healthy, mild and severe patients, where each data point represents a single patient (**E**). Statistics were calculated by one-way ANOVA test (Kruskal-Wallis test) with Dunn's multiple comparison test (**A, E**) and by ANOVA/repeated-measures one-way ANOVA, mixed-effects analysis with the Geisser-Greenhouse correction, Tukey's multiple comparison test. (**B, C**) Data are visualised as mean ± SEM.

The online version of this article includes the following figure supplement(s) for figure 3:

**Figure supplement 1.** Pro-inflammatory cytokines/chemokines in the plasma of coronavirus disease-19 (COVID-19) patients at 3 months post admission.

assay. The ELISpot results showed that the memory T-cell response to SARS-COV-2 spike, membrane and nucleocapsid was robust and comparable in mild, moderate, and severe patients at 3 months post admission. T-cell responses targeting CMV and polyclonal T-cell activation induced by PMA/ionomycin were also similar in mild, moderate, and severe patients (*Figure 4A–F*). The percentage of 'responders' to SARS-CoV-2, calculated as the percentage of patients who presented a response of >5 SFC/10 (*Kuri-Cervantes et al., 2020*) PBMCs to peptides spanning the indicated protein sequence, was higher in moderate versus severe patients but similar between mild and moderate patients (*Figure 4G*). We also assessed the capacity of mild, moderate, and severe patients to mount an antibody response targeting SARS-CoV-2 by using a highly sensitive plate-based luciferase immunoprecipitation assay to the spike receptor binding domain (RBD), which detects all antibody isotypes. SARS-CoV-2 antibody responses were detectable across all patient groups but were higher in patients with moderate compared to mild disease (*Figure 4H*).

In summary, the overall magnitude of IFN-γ producing memory T-cells targeting SARS-CoV-2 spike, membrane and nucleocapsid assessed by ELISpot was comparable between mild, moderate, and severe patients at 3 months post admission. However, the proportion of patients who displayed a T-cell response to SARS-CoV-2 peptides (responders) was higher in moderate compared to severe patients. The titers of RBD-specific antibodies were also higher in moderate compared to mild patients, suggesting a superior capacity of patients with moderate disease to mount detectable memory T-cell and antibody responses to SARS-CoV-2.

## Lower frequencies of IFN-γ+/TNF-α+ and/or CD107a+ SARS-CoV-2-specific T-cells in mild patients

To evaluate the relative contribution of CD4+ and CD8+ T-cells to the SARS-CoV-2 T-cell response observed by ELISpot and to evaluate the cytokine profiles and phenotype of SARS-CoV-2-specific CD4+ and CD8+ T-cells in these patients in-depth, we performed intracellular cytokine staining (ICS) by flow cytometry in samples from 39 patients for which we had available PBMC samples, at 3 months post admission. PBMCs were briefly stimulated with or without SARS-CoV-2 peptides spanning the spike protein or with PMA/ionomycin as a positive control, and cells were subsequently stained with antibodies recognising T-cell markers of differentiation (CD45RA, CCR7), activation (HLA-DR, CD38), proliferation (Ki67), and tissue-homing (CXCR3) and assessed for the production of IFN-γ, TNF-α, and IL-2. Representative flow cytometry plots showing production of IFN-γ, TNF-α, and CD107a are shown in *Figure 5A*. Due to limited cell numbers we were able to comprehensively assess in all samples the T-cell response to only spike-1 peptide pool. Results following spike-2 peptide stimulations were similar to those obtained with spike-1, but were available for fewer patients (data not shown). Our results show that frequencies of spike-1-specific CD4+ memory T-cells producing IFN-γ and/or TNF-α or IFN-γ and/or CD107a upon encounter of SARS-CoV-2 spike-1 peptides were consistently higher in moderate compared to mild patients. In contrast, CD8+ T-cells producing IFN-γ and/or TNF-α were similar across the disease severities and those producing IFN-γ and/or CD107a were higher in moderate and severe compared to mild patients (*Figure 5B*). The polyfunctionality of T-cells, defined as their ability to simultaneously perform more than one function, has been associated with better viral control in chronic viral infections such as HIV. We therefore investigated the polyfunctionality of CD4+ and CD8+ T-cells specific for SARS-CoV-2 spike peptide pools. Across all patient groups monofunctional T-cells performing a single function dominated the response, representing 66–91% of total spike-1-specific T-cells, followed by T-cells performing two or three functions. T-cells performing

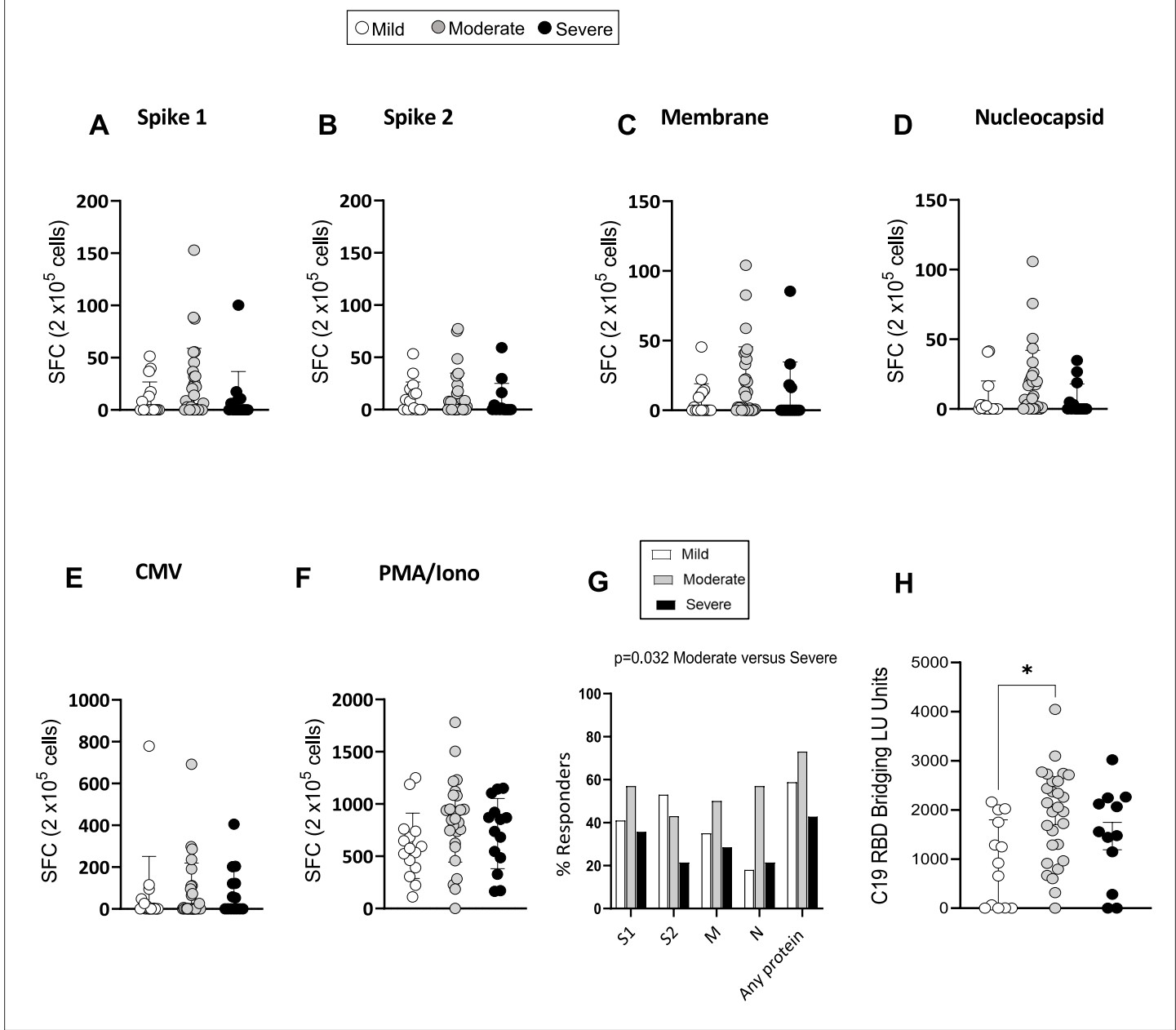

**Figure 4.** Severe acute respiratory syndrome *coronavirus 2* (SARS-CoV-2)-specific memory T-cell and antibody response at 3 months. (**A–F**) Interferon gamma (IFN-γ) release measured by Enzyme-Linked Immune absorbent Spot (ELISpot) in peripheral blood mononuclear cells (PBMCs) from mild, moderate, and severe patients (N=61) upon stimulation with 15-mer peptide pools spanning SARS-CoV-2 spike 1 (**A**), spike 2 (**B**), membrane (**C**), nucleocapsid (**D**), cytomegalovirus (CMV) pp65 (**E**), and PMA/ionomycin (**F**). Results are shown as spot forming cells (SFC) relative to $2×10^5$ PBMCs. (**G**) Percentages of responders assessed as patients from each severity group who displayed a response to the indicated peptide pool >5 SFC/$2×10^5$ PBMCs. (**H**) SARS-CoV-2 receptor binding domain (RBD) antibody titers in patients expressed as RBD bridging LU units. Data in A–F are visualised as mean ± SEM. Statistics were calculated by one-way ANOVA (Kruskal-Wallis test) with Dunn's correction for multiple testing.

four functions (IFN-γ+ IL-2+ TNF-α+ CD107a+) were largely absent (*Figure 5C and D*, and data not shown for IL-2). Across all patient groups, monofunctional CD4+ and CD8+ T-cells targeting spike-1 peptide pools were significantly higher than their polyfunctional counterparts (*Figure 5E*), suggesting that T-cell functionality of SARS-CoV-2 memory T-cells is not influenced by disease severity. Spike-1-specific CD4+ T-cells were mainly contained within the T$_{EM}$ population, while spike-1-specific CD8+ T-cells were contained within T$_{EM}$ and T$_{EMRA}$ populations (*Figure 5F*), and results were similar between

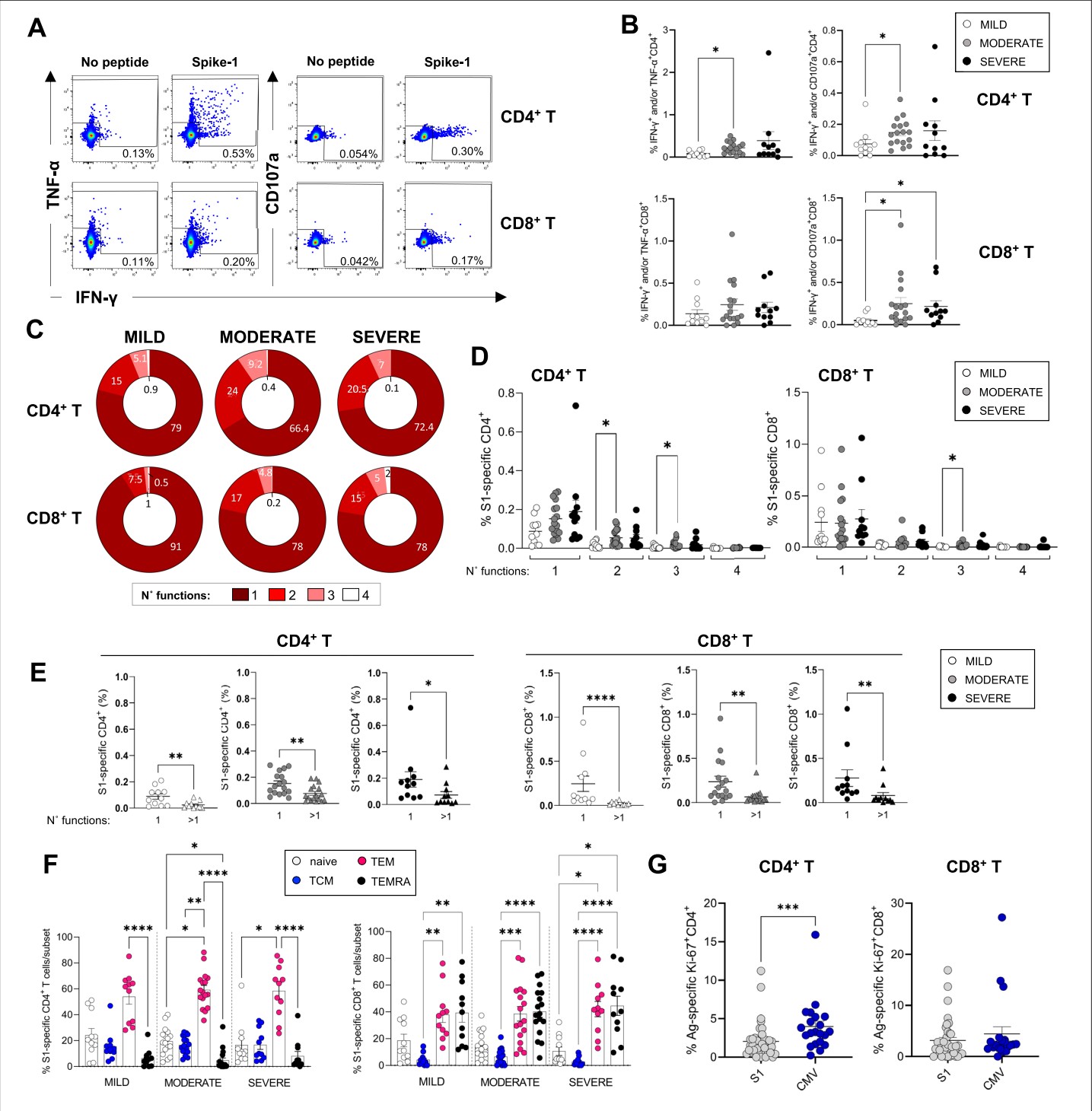

**Figure 5.** Magnitude and cytokine profiles of severe acute respiratory syndrome *coronavirus 2* (SARS-CoV-2)-specific CD4+ and CD8+ T-cells at 3 months. CD4+ and CD8+ T-cell responses targeting spike peptides were assessed by intracellular cytokine staining (ICS) in mild (N=11), moderate (N=17), and severe (N=11) patients. (**A, B**) Shown are representative flow cytometry plots of interferon gamma (IFN-γ) and tumor necrosis factor-alpha (TNF-α) or CD107a production by CD4+ and CD8+ T-cells (**A**) and the percentages of CD4+ (top panel) and CD8+ T (bottom panel) cells producing IFN-γ and/or TNF-α and IFN-γ and/or CD107a in the presence of spike-1 peptides (**B**). (**C**) Pie charts summarising the multifunctionality of T-cells specific for spike-1, defined as their capacity to produce 1, 2, 3, or 4 cytokines/CD107a (no. functions). (**D**) Spike-1 (S1) specific CD4+ (left panel) and CD8+ T-cells (right panel) that express 1–4 functions in mild, moderate, and severe patients. (**E**) Monofunctionality and polyfunctionality (>1 function) of CD4+ and CD8+ T-cells targeting spike-1 peptides in mild, moderate, and severe patients. (**F**) Expression of differentiation markers CD45RA/CCR7 by spike-1specific CD4+ and CD8+ T-cells in mild, moderate, and severe patients. Naïve cells = CCR7+CD45RA+ (white); T central memory cells (T_CM)=CCR7+ CD45RA- (blue);

*Figure 5 continued on next page*

*Figure 5 continued*

T effector memory cells (T$_{EM}$)=CCR7$^+$ CD45RA$^-$ (red); T effector memory RA re-expressing cells (T$_{EMRA}$)=CCR7$^+$ CD45RA$^-$ (black). Data not significantly different between patient groups. (**G**) Percentage of spike-1-specific or CMV-specific CD4$^+$ (left panel) and CD8$^+$ T (right panel) cells that express Ki67. Data in A–B, D–G are visualised as mean ± SEM. Statistics were calculated by one-way ANOVA (Kruskal-Wallis test) with Dunn's correction for multiple testing or by Mann-Whitney t-test.

patient groups. CXCR3 expression was also similar in CD4$^+$ and CD8$^+$ T-cells across patient groups (data not shown).

We next asked whether we could detect any ongoing activation and/or proliferation of SARS-CoV-2 or CMV-specific T-cells at 3 months post admission to understand whether these cells contributed to the total pool of activated/proliferating cells that were increased in severe patients. Our results showed that CMV-specific CD4$^+$ but not CD8$^+$ T-cells expressed Ki67 at significantly higher levels compared to their spike-1-specific counterparts, suggesting that CMV-specific CD4$^+$ T-cells were proliferating albeit at low levels (*Figure 5G*). These data suggest that CMV-specific CD4$^+$ T-cells may contribute to the pool of activated CD4$^+$ T-cells detected in peripheral blood of COVID-19 patients at 3 months.

In summary, SARS-CoV-2-specific CD4$^+$ and CD8$^+$ T-cells could be detected in the peripheral blood of patients 3 months after mild, moderate, and severe COVID-19. An in-depth analysis of SARS-CoV-2-specific T-cells producing IFN-γ, TNF-α, IL-2, and CD107a revealed an increased frequency of T-cells targeting spike-1 in moderate and/or severe compared to mild patients.

## Associations between ongoing symptoms and immune response

At the 3 month follow-up clinic, patients were screened for a pre-defined list of symptoms in a clinician-led clinic, as reported elsewhere (*Arnold et al., 2021*). Patients also completed a 36-item Short Form Survey (SF-36), and physical component summary (PCS) and mental component summary (MCS) scores were calculated (*Ware and Kosinski, 2001*; *Brazier et al., 1992*). Of all patients included in this study, 79% (82%, 75%, and 86% of mild, moderate, and severe patients, respectively) reported at least one ongoing symptom with breathlessness and excessive fatigue being the most common. Symptoms also included muscle weakness, sleeping difficulty, psychiatric symptoms, anosmia, chest pain, and cough (*Table 1*). Patients who recovered from severe disease experienced each of these symptoms more frequently compared to patients from the mild and moderate groups, except for cough and chest pain which were reported by a minority of patients across all groups. The proportion of patients who did not report any symptoms was largely similar in mild, moderate, and severe patients (*Figure 6A*). Mild patients more frequently reported 1 symptom, while those from the severe group were more likely to report 4 symptoms. Moderate patients reported 0–2 symptoms with similar frequencies (*Figure 6B*). SF-36 PCS and MCS scores at 3 months were similar between mild, moderate, and severe patients (data not shown).

We next asked whether ongoing symptoms, PCS scores or MCS scores at 3 months associate with T-cell profiles and/or the plasma pro-inflammatory cytokines assessed in this study. Poisson regression was performed for these outcome measures against a set of immune parameters of interest including cytokines/chemokines that could be detected consistently across patients (IFN-γ, IL-12p70, IL-13, IL-15, IL-17A, IL-18, IL-4, IL-7, IP-10, and TNF-α) and T-cell profiles detected by flow cytometry (populations identified by manual gating and UMAP/FlowSOM). In the analyses that were unadjusted for sex, age, and severity the number of long COVID symptoms was found to associate significantly with two clusters of T-cells that were identified in the FlowSOM analyses (CD4$^+$ T-cell cluster 2: FDR p=0.003 and CD8$^+$ T-cell cluster 4: FDR p=0.015, *Figure 6C and D*, respectively; *Figure 6—source data 1A*). CD4$^+$ T-cell cluster 2 represents a population of cells present at higher frequencies in severe compared to moderate patients, which expressed CCR7, CD38, and intermediate/low levels of Ki67, suggesting moderate/recent activation (*Figure 2N*). These cells represent on average approximately 50% of all CD4$^+$ T-cells in severe patients. In contrast, CD8$^+$ T-cell cluster 4 is present in similar frequencies across disease severities and represents a subset of cells that express HLA-DR, intermediate levels of CCR7 and CXCR3 and lacks granzyme B expression (*Figure 3K*). Other immune parameters including the frequencies of activated/proliferating CD4$^+$ T-cell subsets and UMAP CD4$^+$ T-cell cluster 11 (CD38 high, Ki67 intermediate CD4$^+$ T-cells) associated inversely with the number of symptoms with uncorrected p-values <0.05, but these associations were not statistically significant after FDR correction. The ratio of CD4$^+$/CD8$^+$ T-cells correlated directly with the number of symptoms but similarly

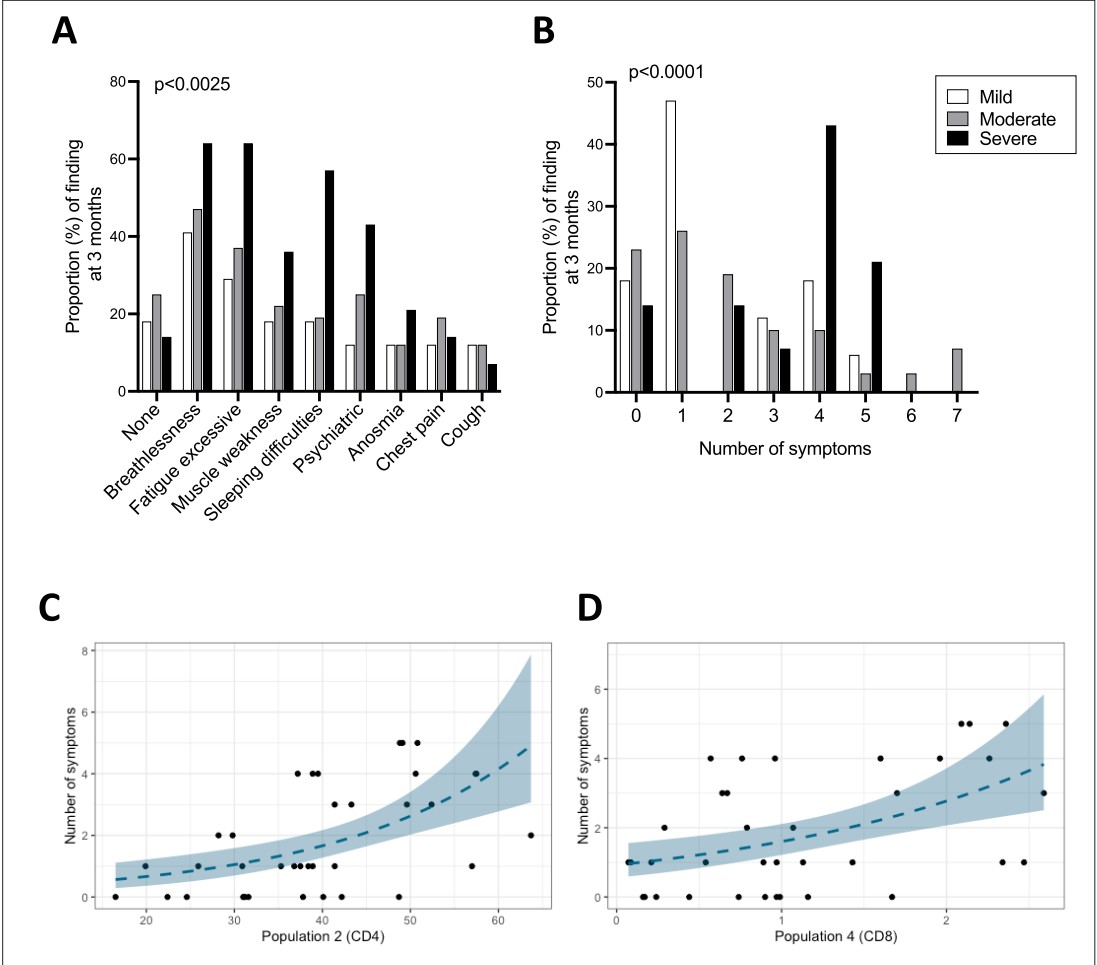

**Figure 6.** Ongoing symptoms at 3 months and associations with immune profiles. (**A**, **B**) The percentage of patients with mild (N=17), moderate (N=32), and severe (N=14) coronavirus disease-19 (COVID-19) who reported the indicated symptom (**A**) or number of symptoms (**B**) at 3 months are indicated with white, grey, and black bars, respectively. Statistics were calculated using a Chi-square test. (**C, D**) Graphs depicting the association between number of symptoms and uniform manifold approximation and projection (UMAP) T-cells clusters in Poisson models, specifically CD4+ T-cell cluster 2 (**C**) and CD8+ T-cell cluster 4 (**D**).

The online version of this article includes the following source data for figure 6:

**Source data 1.** Associations between immune parameters and symptoms, physical component summary (PCS) or mental component summary (MCS) scores at 3 months in either unadjusted (**A**) or adjusted (**B**) Poisson regression models.

the association was not significant after FDR correction (*Figure 6—source data 1A*). In an analysis adjusted for sex, age, and severity none of the estimates displayed FDR p<0.05 (*Figure 6—source data 1B*). FlowSOM CD4+ T-cell cluster 2 and CD8+ T-cell cluster 4 that significantly associated with the number of symptoms in the unadjusted analysis failed to associate significantly after adjusting for age, sex, and severity. Other immune parameters including plasma cytokine levels of IP10, IL-4, IFN-γ, and IL-12, UMAP CD8+ T-cell cluster 8 and frequencies of Ki67+ HLA-DR+ CD4+ $T_{EM}$ cells directly correlated with number of symptoms (p<0.05), but p-values were >0.05 after FDR correction (*Figure 6—source data 1B*). There were no significant associations between PCS or MCS scores and T-cell profiles or cytokines/chemokines in either unadjusted or adjusted analyses after FDR correction.

## Discussion

Emerging data demonstrates that immune activation may persist for months after COVID-19, however there is heterogeneity in findings particularly between hospitalised versus non-hospitalised cohorts, which by nature of how patients were managed during the peak of the pandemic comprise of patients

with a very different spectrum of disease. Here, we report results from a small study performed on a cohort of clinically well-characterised mild, moderate, and severe patients who were all hospitalised at the start of the pandemic and followed up closely through outpatient follow-up clinics up to 12 months post admission. We ask whether ongoing inflammation and immune perturbations, particularly within the T-cell population, associate with long COVID symptoms. Specifically, we ask the following questions: (1) Is there evidence of ongoing inflammatory events post hospitalisation with COVID-19, and are these linked with disease severity at acute infection? (2) Are the magnitude and features of the SARS-CoV-2-specific CD4$^+$ and CD8$^+$ T-cell memory response influenced by disease severity? (3) Do long COVID symptoms associate with persistent immune activation, inflammation, and/or SARS-CoV-2-specific T-cell immunity? To answer these questions, we performed an in-depth immunological analysis of 187 samples from 63 patients at 3 and 12 months post admission, including a characterisation of activation/proliferation profiles and phenotypic features of circulating CD4$^+$ and CD8$^+$ T-cells, γδ-T-cells, NK cells, B cells, and CD14$^{+/-}$CD16$^{+/-}$ monocytes by flow cytometry, as well as an analysis of T-cells and antibodies targeting SARS-CoV-2 and pro-inflammatory plasma cytokines/chemokines by Luminex at 3, 8, and 12 months. Our data shows persistent CD4$^+$ and CD8$^+$ T-cell activation and elevated plasma levels of IL-4, IL-7, IL-17, and TNF-α at 3 months, in severe compared to mild and/or moderate patients. The elevated levels of these cytokines are consistent with ongoing T-cell activation as CD4$^+$ T-cells are key producers of IL-4 and IL-17 and both CD4$^+$ and CD8$^+$ T-cells secrete TNF-α. IL-7 stimulates proliferation of T-cells and can be produced by a variety of cells such as epithelial cells, stromal cells, and dendritic cells. Plasma levels of IL-4, IL-7, and IL-15 were similar at acute infection and 3 months but declined at later time points, suggesting a slow recovery of T-cell-related perturbations within 12 months. CD4$^+$ and CD8$^+$ T-cell activation was evident when we analysed the flow cytometry data both by manual gating and by FlowSOM/UMAP - where the former method investigates cells expressing known combinations of markers, while the latter identifies clusters of cells in an unsupervised manner based on the combined expression of all markers analysed. Manual gating highlighted increased levels of CD4$^+$ T-cells expressing the lung tissue-homing marker CXCR3 in severe patients, suggesting the presence of factors that may be recruiting and priming CD4$^+$ T-cells in this site. This finding may suggest unresolved damage/inflammation in the lung, however clinical abnormalities in lung function were evident only in a minority of these patients during the follow-up clinic (**Al-Aly et al., 2022**). Both manual and FlowSOM/UMAP analysis show increased granzyme B expression in CD8$^+$ T-cells from severe patients, suggesting an overall higher cytotoxic ability of these cells. Granzyme B expression is upregulated in cells after T-cell receptor (TCR) stimulation and confers higher cytotoxic abilities to the activated T-cell, hence higher expression could reflect a higher activation state in these cells. Interestingly, expression of CXCR3 is reduced (and undetectable) in both CD4$^+$ and CD8$^+$ T-cells at 12 compared to 3 months post admission (**Figure 2—figure supplement 1**). Expression of granzyme B was also decreased in CD8$^+$ T-cells at 12 compared to 3 months. Ongoing T-cell activation did not appear to impair the ability of severe patients to mount and maintain memory T-cell or antibody responses to SARS-CoV-2 structural proteins, which were present and robust across all patients. The quality of the T-cell response was also similar across the patient groups, and we could detect monofunctional and polyfunctional SARS-CoV-2-specific T-cells, with the former dominating the response. However, moderate compared to mild patients displayed elevated spike RBD antibody titers and higher magnitudes of CD4$^+$ and CD8$^+$ T-cells producing IFN-γ and/or TNF-α/CD107a by ICS, after stimulation with spike-1 peptides. The magnitude of the T-cell response is closely linked with viremia (**Miller et al., 2008**) and higher viral loads during the acute illness may underlie the higher magnitude of IFN-γ/TNF-α producing CD4$^+$ and CD8$^+$ T-cells. However, viremia data at acute illness was not available for our patient cohort. The frequency of T-cells producing IFN-γ alone did not differ in the ICS analysis, in line with the ELISpot data. Cytokine production following PMA/ionomycin were comparable between patients, suggesting similar T-cell functionality to polyclonal stimulation that bypasses the TCR.

T-cells can be activated through TCR triggering or 'bystander activated' by cytokines such as IL-7 and IL-15. Bystander T-cell activation was observed to occur during viral infection and is likely driven by IL-15 (**Sandalova et al., 2010**; **Rivino et al., 2015**). Here, we demonstrate that co-culture of plasma from severe patients induced upregulation of the IL-15 receptor (IL-15R)-α-chain on T-cells from healthy donors, suggesting that the low-level proliferation of T-cells we see in severe patients may be bystander in nature and driven by plasma cytokines persistently present at 3 months in severe

patients. Highly differentiated T-cells are more responsive to IL-15 due to their higher expression of the IL-15R (*Geginat et al., 2003*; *Rivino et al., 2004*). The higher responsiveness of CMV-specific T-cells, which are highly differentiated cells, has been proposed as a mechanism to explain why these cells are commonly activated/proliferating during acute viral infections such as dengue, HBV, and COVID-19 (*Sandalova et al., 2010*; *Rivino et al., 2015*; *Gregorova et al., 2020*). It may be possible that the low-level proliferation of CMV-pp65-specific CD4$^+$ T-cells observed in this study is driven by cytokines which are still elevated at 3 months. Our data does not support the persistence of SARS-CoV-2 antigens at 3 months for two reasons. Firstly, we would expect that the persistence of SARS-CoV-2 antigens in vivo would result in the activation of SARS-CoV-2-specific T-cells, however we did not observe expression of markers of activation nor proliferation in spike-1-specific T-cells in our ICS analysis. Secondly, we did not detect any SARS-CoV-2 RNA by RT-PCR - however we cannot exclude that SARS-CoV-2 antigens may still be present but are sequestered in specific sites to which T-cells do not have access.

Prolonged T-cell activation after acute infection has been observed in other severe respiratory infections and may not be specific for COVID-19. In H7N9 influenza infection persistent T-cell activation was associated with fatal disease while T-cell activation early during infection associated with positive clinical outcomes, however the frequencies of bystander-activated T-cells in fatal patients was high (approx. 20% of CD8$^+$ T-cells) (*Wang et al., 2018*). Further studies are needed to understand the impact of prolonged T-cell activation following acute viral infections.

In this study the levels of CRP, albumin, and IL-6 at 3 months were similar across patients, suggesting an overall resolution of the inflammatory processes occurring during the acute infection. In contrast, a recent study showed increased levels of several pro-inflammatory markers at 12 months, including CRP and IL-6, which correlated with the presence of persistently activated T-cells. Interestingly the authors showed that COVID-19 mRNA vaccination and boosting of SARS-CoV-2 immunity did not alter these profiles, suggesting inflammation may be independent of the presence of SARS-CoV-2 antigens in these patients (*Taeschler et al., 2022*). We cannot exclude that the presence of activated T-cells may be linked to the comorbidities experienced by the patients in this cohort, which included obesity, type-1 and -2 diabetes, and asthma. These comorbidities were present across all groups but increased progressively with disease severity (*Table 1*). However, at 12 months post admission T-cell activation was similar and largely undetectable in patients with mild, moderate, and severe disease, suggesting that the differences we observe between patients at 3 months may not be driven by differences in T-cell activation already present in these patients prior to COVID-19.

Due to the limited number of cells available we were unable to analyse T-cell phenotypes associated with regulatory and anti-inflammatory activity. This represents a limitation of this study. The relatively small patient cohort as well as the lack of ethnic diversity of patients who were predominately white Caucasians represents another limitation of this study.

Long COVID symptoms were reported in 80% of patients across all disease severities but were more frequent in patients with severe disease, in line with published findings (*Cabrera Martimbianco et al., 2021*). The factors driving long COVID remain largely unclear, as do those underlying the long-term fatigue syndromes that have previously been observed following other viral infections (*Hickie et al., 2006*). Although the prevalence of long COVID is reported to be higher in hospitalised patients and those who received intensive care support, the risk factors of developing a severe acute illness (male sex, age, obesity, ethnicity, and cardiovascular disease) only partially overlap with those reported to associate with long COVID (female sex, age, obesity, anxiety, asthma), suggesting that pathogenesis at acute infection and in convalescence may be driven by distinct mechanisms (*Evans et al., 2022*; *Sneller et al., 2022*; *Halpin et al., 2021*; *Thompson et al., 2022*). In this study, Poisson regression analysis showed no association between ongoing symptoms and immune perturbations after adjusting for age, sex, and severity grades, but identified a significant association between ongoing symptoms and the frequency of a cluster of moderately activated CD4$^+$ T-cells (cluster 2) and a cluster of activated, non-cytotoxic HLA-DR$^+$ CD8$^+$ T-cells (cluster 4) in unadjusted analyses, after FDR correction. Our study adds to emerging data suggesting that a prolonged immune activation can be observed following COVID-19 which may not directly associate with long COVID (*Evans et al., 2022*; *Sneller et al., 2022*). The clinical implications of persistent T-cell activation following COVID-19 remain unclear and warrant further investigation.

In summary, our study highlights a complex recovery of the immune system following severe COVID-19 with evidence of persistent activation of CD4[+] and CD8[+] T-cells, which may be bystander driven, and elevated levels of T-cell-related cytokines in the plasma of severe patients at 3 months. Our data suggests the lack of a direct association between long COVID and immune activation markers and pro-inflammatory cytokines measured in this study.

# Materials and methods

**Key resources table**

| Reagent type (species) or resource | Designation | Source or reference | Identifiers | Additional information |
|---|---|---|---|---|
| Antibody | Mouse monoclonal anti-human CD4 (RPA-T4) | Biolegend | Cat# 300535 | FC (0.625:50) |
| Antibody | Mouse monoclonal anti-human HLA-DR (L243) | Biolegend | Cat# 307640 | FC (2.5:50) |
| Antibody | Mouse monoclonal anti-human CD38 (HIT2) | Biolegend | Cat# 303528 | FC (2.5:50) |
| Antibody | Mouse monoclonal anti-human Ki-67 (Ki-67) | Biolegend | Cat# 350505 | FC (3:50) |
| Antibody | Mouse monoclonal anti-human CD16 (3G8) | Biolegend | Cat# 302007 | FC (2.5:50) |
| Antibody | Mouse monoclonal anti-human CD8 (SK1) | Biolegend | Cat# 344713 | FC (3:50) |
| Antibody | Mouse monoclonal anti-human CD56 (NCAM16.2) | BD Biosciences | Cat# 564849 | FC (0.5:50) |
| Antibody | Mouse monoclonal anti-human CD3 (UCHT1) | BD Biosciences | Cat# 557943 | FC (1:50) |
| Antibody | Mouse monoclonal anti-human IFN-γ (B27) | BD Biosciences | Cat# 560371 | FC (3:50) |
| Antibody | Mouse monoclonal anti-human CD3 (UCHT1) | BD Biosciences | Cat# 560835 | FC (2:50) |
| Antibody | Mouse monoclonal anti-human CD107a (H4A3) | Biolegend | Cat# 328610 | FC (0.75:50) |
| Antibody | Mouse monoclonal anti-human TNF-α (MAb11) | Biolegend | Cat# 502946 | FC (3:50) |
| Antibody | Mouse monoclonal anti-human CD163 (GHI/61) | Biolegend | Cat# 333631 | FC (2.5:50) |
| Antibody | Mouse monoclonal anti-human CD14 (M5E2) | Biolegend | Cat# 301833 | FC (2.5:50) |
| Antibody | Mouse monoclonal anti-human CD68 (Y1/82A) | Biolegend | Cat# 333811 | FC (5:50) |
| Antibody | Mouse monoclonal anti-human CD66b (G10F5) | Biolegend | Cat# 305122 | FC (5:50) |
| Antibody | Mouse monoclonal anti-human TCR γ/δ (B1) | Biolegend | Cat# 331209 | FC (2.5:50) |
| Antibody | Mouse monoclonal anti-human CD16 (3G8) | Biolegend | Cat# 302017 | FC (0.5:50) |
| Antibody | Mouse monoclonal anti-human CD80 (2D10) | Biolegend | Cat# 305219 | FC (2.5:50) |
| Antibody | Mouse monoclonal anti-human CD69 (FN50) | Biolegend | Cat# 310931 | FC (5:50) |
| Antibody | Mouse monoclonal anti-human CXCR3 (G025H7) | Biolegend | Cat# 353714 | FC (1:50) |

*Continued on next page*

*Continued*

| Reagent type (species) or resource | Designation | Source or reference | Identifiers | Additional information |
|---|---|---|---|---|
| Antibody | Mouse monoclonal anti-human CD45RA (HI100) | BD Biosciences | Cat# 561882 | FC (10:50) |
| Antibody | Mouse monoclonal anti-human CCR7 (G043H7) | Biolegend | Cat# 353226 | FC (2.5:50) |
| Antibody | Mouse anti-human CD19 (HIB19) | Biolegend | Cat# 302216 | FC (1:50) |
| Antibody | Rat anti-human IL-2 (MQ1-17H12) | Biolegend | Cat# 500322 | FC (2:50) |
| Antibody | Mouse anti-human CD8a (RPA-T8) | Biolegend | Cat# 301014 | FC (0.3:50) |
| Antibody | Mouse anti-human IL-15Rα (JM7A4) | Biolegend | Cat# 330207 | FC (2.5:50) |
| Antibody | Mouse monoclonal anti-human PD1 (EH12.1) | BD Biosciences | Cat# 612791 | FC (3:50) |
| Antibody | Mouse recombinant anti-human Granzyme B (QA16A02) | Biolegend | Cat# 372219 | FC (2.5:50) |
| Biological sample (human) | Peripheral blood mononuclear cells (PBMCs) | DISCOVER study, Bristol, UK | | Frozen - isolated PBMCs |
| Biological sample (human) | Plasma | DISCOVER study, Bristol, UK | | Frozen plasma |
| Peptide, recombinant protein | SARS-CoV-2 spike protein overlapping peptide library (custom made) | Mimotopes | N/A | |
| Peptide, recombinant protein | SARS-CoV-2 membrane protein overlapping peptide library (custom made) | Mimotopes | N/A | |
| Peptide, recombinant protein | SARS-CoV-2 nucleocapsid protein overlapping peptide library (custom made) | Mimotopes | N/A | |
| Peptide, recombinant protein | CMV pp65 protein (AD169 strain) overlapping peptide library (custom made) | Mimotopes | N/A | |
| Other | Spike-RBD Antibody Bridging LIPS assay | DOI: 10.3389/fimmu.2022.968317 | N/A | |
| Peptide, recombinant protein | Nano-Glo | Promega | Cat# N1150 | |
| Commercial assay or kit | Human IFN-γ ELISpot BASIC kit | Mabtech | Cat# 34202A | |
| Commercial assay or kit | ProcartaPlex Mix&\Match 23-plex | Invitrogen | Cat# PPX-23-MXWCXFA | |
| Commercial assay or kit | Pan T Cell Isolation Kit, human | Miltenyi Biotec | Cat# 130-096-535 | |
| Commercial assay or kit | CellTrace Violet Cell Proliferation Kit, for flow cytometry | Thermo Fisher Scientific | Cat# C34557 | |
| Commercial assay or kit | eBioscience Foxp3/Transcription factor fixation/permeabilisation buffer | Invitrogen | Cat# 00-5523-00 | |
| Commercial assay or kit | Zombie Aqua Fixable Viability Kit | Biolegend | Cat# 423102 | FC (1:1000) |
| Commercial assay or kit | Zombie NIR Fixable Viability Kit | Biolegend | Cat# 423105 | FC (1:100) |
| Commercial assay or kit | Dynabeads Human T-Activator CD3/CD28 | Thermo Fisher Scientific | Cat# 11131D | |

*Continued on next page*

*Continued*

| Reagent type (species) or resource | Designation | Source or reference | Identifiers | Additional information |
|---|---|---|---|---|
| Commercial assay or kit | OneComp eBeads Compensation Beads | Thermo Fisher Scientific | Cat# 01-1111-42 | |
| Commercial assay or kit | Human TruStain FcX | Biolegend | Cat# 422302 | FC (2.5:50) |
| Commercial assay or kit | Human Anti-Cytomegalovirus IgG ELISA Kit (CMV) | Abcam | Cat# ab108724 | |
| Software | FlowJo | BD | v10.8.1 | |
| Software | R | R Foundation for Statistical Computing | v4.0.4 | |
| Software | GraphPad Prism | GraphPad Software | v9.4 | |
| Software, algorithm | xPONENT | Software for Luminex Instruments | The basic xPONENT software | |
| Software, algorithm | BioSpot Software Suite | ImmunoSpot S6 Ultra-V Analyzer | | |

## Patients

Patients hospitalised with COVID-19 (≥18 years of age) were recruited between 30 March and 3 June 2020 into the observational study DIagnostic and Severity markers of COVID-19 to Enable Rapid triage (DISCOVER), a single-centre prospective study based in Bristol (UK). Research Ethics Committee (REC) approval: REC:20/YH/1021. Survivors were invited at 3, 8, and 12 months post admission to attend outpatient follow-up clinics for a systematic clinical assessment (*Arnold et al., 2021*). For those patients attending a face-to-face follow-up, consent was taken to collect samples for research purposes (blood for PBMC isolation, plasma, and serum). When available serum collected from patients at admission was made available to the research team.

## RT-PCR analysis

RT-PCR analysis was performed on all plasma samples used for Luminex for biosafety reasons (acute, 3–12 months). SARS-CoV-2 test was performed by an in-house RT-PCR at the regional Southwest Public Health England Regional Virology laboratory, utilising a PHE-approved assay at the time of testing. All plasma samples were RT-PCR negative except for two samples taken at acute infection.

## Isolation of PBMCs

Blood samples were collected from COVID-19 patients after informed consent in EDTA vacutainer tubes and PBMCs and plasma were isolated from peripheral blood using Leucosep tubes containing Ficoll and cryopreserved. PBMCs were resuspended in freezing media (10% DMSO, 90% FCS) prior to freezing in –80°C and frozen vials were transferred to liquid nitrogen 24–48 hr later.

## Flow cytometry staining and PBMC stimulation

PBMCs were thawed and either stained ex vivo or stimulated in AIMV 2% FCS with or without peptide pools from SARS-CoV-2 spike, membrane (M), nucleocapsid (N), CMV pp65 (all 1 µg/ml), or with PMA/ionomycin (PMA 10 ng/ml, ionomycin 100 ng/ml, Sigma-Aldrich) for 5 hr at 37°C in the presence of brefeldin A (BD, 5 µg/ml). To assess degranulation, CD107a antibody was added to the cells at the beginning of the stimulation. Cells were stained with a viability dye Zombie Aqua (Biolegend) for 10 min at room temperature and then with antibodies targeting surface markers (20 min 4°C, diluted in PBS 1% BSA [Sigma-Aldrich]). Cells were fixed overnight in eBioscience Foxp3/Transcription factor fixation/permeabilisation buffer (Invitrogen), and intracellular staining was performed for detection of Ki67, granzyme B, or intracellular cytokines (30 min 4°C). Cells were acquired on a BD LSR Fortessa X20 and data analysed using FlowJo software v10.8.1. A complete list of antibodies is included in Key resources table.

## Flow cytometry data analysis

Flow cytometry data was analysed in parallel by manual gating methods and by using unsupervised multi-dimensional algorithms. For the latter analysis we concatenated the flow cytometry standard (FCS) files containing the data from 56 patient samples and performed a FlowSOM clustering analysis and visualised identified clusters using UMAP for different cell populations (e.g., CD3+, CD4+, and CD8+). FCS files from 56 patients were concatenated after downsampling and UMAP and FlowSOM analyses were done by using a plugin in FlowJo v10.8.1. All FCS files for the experiments included in this paper can be accessed in FlowRepository. Repository IDs: FR-FCM-Z5VC (T/NK-cell phenotyping); FR-FCM-Z5VB (innate cell phenotyping); FR-FCM-Z5VA (ICS); FR-FCM-Z5VD and FR-FCM-Z5VE (IL-15R co-culture experiments).

## T-cell co-culture with patient plasma

PBMCs were isolated from the blood of healthy donors (N=4) and CD3+ T-cells were isolated with magnetic beads using Pan T-cell Isolation Kit (Miltenyi Biotec), according to the manufacturer's instructions. Purified CD3+ T-cells from healthy donors were then labelled with CellTrace Violet (Thermo Fisher Scientific) and $3 \times 10^5$ purified T-cells were co-cultured in round-bottom 96-well plate with plasma derived from either a heterologous healthy donor (N=4 healthy plasma), severe (N=4 severe plasma), or mild patients (N=4 mild plasma) for 7 days. Each condition was performed in technical duplicates. Cells were also plated in the presence of anti-CD3/CD28 Dynabeads (Thermo Fisher Scientific) as a positive control. Following the incubation period, cells were stained with Zombie NIR (Biolegend) for 10 min at room temperature before staining cells for 20 min at 4°C with a cocktail of the following antibodies diluted in PBS 1% BSA (Sigma-Aldrich): anti-CD4 BV650, anti-CD8 APC, anti-CD215(IL-15Rα) PE, anti-CD3 Percy5.5. Details of the antibodies are reported in Key resources table. Cells were acquired on a BD LSR Fortessa X20 cytometer.

## Synthetic peptides

15-mer peptides overlapping by 10 amino acid residues and spanning the SARS-CoV-2 spike, membrane and nucleocapsid protein and CMV pp65 (AD169 strain) proteins were purchased from Mimotopes (Australia). The purity of the peptides was >80% or >75%, respectively. Peptides were dissolved as described previously (*Rivino et al., 2013*).

## ELISpot

Human IFN-γ ELISpot assays were performed using a human IFN-γ ELISpot BASIC kit (Mabtech). MSIP4W10 PVDF plates (Millipore) were prepared by pre-coating wells in 35% ethanol for 30 s and washing them thoroughly with sterile water to remove any residual ethanol. Subsequently the plates were coated with capture antibody (mAb-1-D1K; 15 µg/ml) diluted in PBS and incubated overnight at 4°C. Cryopreserved PBMCs were thawed and rested at 37°C, 5% $CO_2$ for 5–6 hr. Coated plates were washed five times in sterile PBS and blocked for 1–2 hr using R10 medium which is composed of 0.2 µm filtered RPMI 1640 medium supplemented with 10% FCS, 2 mM glutamine, penicillin (100 units/ml), and streptomycin (100 µg/ml). $2 \times 10^5$ PBMCs were added to each 96-well plate with or without peptide pools (as indicated) in a total volume of 100 µl in R10. PBMCs incubated with R10 medium alone were used as negative (unstimulated) controls. Peptide pools spanning spike (S1, S2, S3, and S4), nucleocapsid (N1 and N2), membrane protein (M), and CMV pp65 were used at a final concentration of 2 µg/ml. PBMC stimulated with PMA at 1 µg/ml and ionomycin at 10 µg/ml (Sigma-Aldrich) as a positive control. All conditions (positive and negative controls and peptide stimulated wells) were performed in duplicate. Plates were incubated for 16–18 hr at 37°C, 5% $CO_2$ and developed as per manufacturer's instructions. Developed plates were protected from light and dried for 24–48 hr before image acquisition using CTL ImmunoSpot S6 Ultra-V Analyzer. All plates were read using the same settings. Spot forming units for each peptide pool were calculated after subtraction of average background calculated from negative control wells. Negative values after background subtraction were adjusted to zero. Responses were considered antigen-specific when spot counts after background subtraction of relevant peptide pool exceeded 2× standard deviations of the unstimulated control wells, as described (*Swadling et al., 2022*).

## Cytokine analysis

A customised Luminex assay was used to measure the following 23 analytes (cytokines/chemokines) in patients' plasma samples: GM-CSF, IFN-α and IFN-γ, IL-1α, IL-1β, IL-2, IL-4, IL-5, IL-6, IL-7, IL-8

(CXCL8), IL-10, IL-12p70, IL-13, IL-15, IL-17A (CTLA-8), IL-18, IP-10 (or CXCL10), MCP-1 (or CCL2), MCP-4, MIP-1α, MIP-1β, and TNF-α. The ProcartaPlex Multiplex Immunoassay kit (Thermo Fisher Scientific, PPX-23-MXWCXFA) was used according to the manufacturer's recommendations. Samples were acquired on Luminex200 Analyser Instrument using the xPONENT basic plus partner analytics. Data were analysed using Invitrogen ProcartaPlex Analysis App on the Thermo Fisher Connect Platform. The two plasma samples from the acute time points that were RT-PCR positive for SARS-CoV-2 were heat inactivated for 30 min at 56°C prior to their use in the Luminex assay. For the matching 3- to 12-month plasma samples, Luminex was performed in parallel on samples that had either been heat inactivated or not, and measurements of all detectable analytes were comparable in these two conditions. Only the data from the non-heat inactivated 3- to 12-month samples was included in *Figure 3*.

## Spike-RBD antibody bridging LIPS assay

This assay was performed as described previously (*Halliday et al., 2022*). Briefly, spike antigen diluted to 2.5 ng/μl in 40 μl PBS was pipetted into every well of a 96-well high-binding OptiPlate (Perkin-Elmer, Waltham, MA, USA) and incubated for 18 hr at 4°C. The plate was washed four times with 20 mM Tris 150 mM NaCl pH 7.4 with 0.15% vol/vol Tween-20 (TBST) and blocked with 1% Casein in PBS (Thermo Scientific, Waltham, MA, USA). The plate was left to air-dry for 2–3 hr before being stored with a sachet of desiccant in a sealed plastic bag at 4°C and used within 3 weeks. The Nluc-RBD antigen was diluted in TBST to $10 \times 10^6 \pm 5\%$ LU per 25 μl. Sera (1.5 μl, 2 replicates) were pipetted into a 96-well plate and incubated with 37.5 μl diluted labelled antigen for 2 hr. Of this mixture, 26 μl was transferred into the coated OptiPlate and incubated shaking (~700 rpm) for 1.5 hr. The plate was washed eight times with TBST, excess buffer was removed by aspiration, then 40 μl of a 1:1 dilution of Nano-Glo substrate (Promega), and TBST was injected into each well before counting in a Hidex Sense Beta Luminometer (Turku, Finland). Units were interpolated from LU through a standard curve.

## Statistical analysis

Statistical analysis was performed using either GraphPad Prism v9.4 (GraphPad Software, San Diego, CA, USA) or R version 4.0.4. The statistical tests and post hoc corrections for multiple testing used are indicated in each figure legend. p-Values are indicated as follows: $*p<0.05$, $**p\leq0.01$, $***p\leq0.001$. For comparison of the immune data and long COVID symptoms, Poisson regression was used with number of symptoms, PCS scores or MCS scores at 3 months as the outcome variable, and each individual cellular marker/cytokine as the explanatory variable. Analyses were performed unadjusted and adjusted for age and sex. Due to the number of comparisons, FDR correction was performed at a value of 5%. Analysis was performed using the tidyverse package. The script and data are publicly available on GitHub (copy archived at *Hamilton, 2022*).

# Acknowledgements

The authors wish to acknowledge the assistance of Dr Andrew Herman, Helen Rice, Poppy Miller, and the University of Bristol Faculty of Biomedical Sciences Flow Cytometry Facility as well as Dr Kapil Gupta at the School of Biochemistry for spike-RBD protein and members of the Diabetes and Metabolism team for help in testing samples for spike-RBD antibodies. We would also like to thank the Bristol UNCOVER team for helpful discussions during the execution of this work and preparation of the manuscript. We are grateful to all patients and their families for participating in this study. This work was supported by the Elizabeth Blackwell Institute (EBI) with funding from the University's alumni and friends (TRACK award to LR and grants to LW and AG) and by Southmead Hospital Charity (Registered Charity Number: 1055900).

## Additional information

### Funding

| Funder | Grant reference number | Author |
|---|---|---|
| Wellcome Trust | Elizabeth Blackwell Institute (EBI) with funding from the University's alumni and friends | Laura Rivino<br>Anu Goenka<br>Linda Wooldridge |
| Southmead Hospital Charity | DISCOVER | Laura Rivino<br>Fergus Hamilton<br>David Arnold |

The funders had no role in study design, data collection and interpretation, or the decision to submit the work for publication. For the purpose of Open Access, the authors have applied a CC BY public copyright license to any Author Accepted Manuscript version arising from this submission.

### Author contributions

Marianna Santopaolo, Michaela Gregorova, Formal analysis, Investigation, Writing - review and editing; Fergus Hamilton, Formal analysis, Funding acquisition, Investigation; David Arnold, Funding acquisition, Investigation; Anna Long, Investigation, Methodology; Aurora Lacey, Kristy Hamilton, Rachel Milligan, Begonia Morales Aza, Formal analysis, Investigation; Elizabeth Oliver, Lea Knezevic, Alice Milne, Emily Milodowski, Investigation; Alice Halliday, Holly Baum, Olivia Pearce, Formal analysis, Investigation, Methodology; Eben Jones, Formal analysis, Writing - review and editing; Rajeka Lazarus, Resources, Writing - review and editing; Anu Goenka, Linda Wooldridge, Supervision, Funding acquisition, Writing - review and editing; Adam Finn, Supervision, Funding acquisition; Nicholas Maskell, Resources, Supervision, Funding acquisition; Andrew D Davidson, Formal analysis, Supervision, Investigation; Kathleen Gillespie, Resources, Formal analysis, Supervision, Methodology; Laura Rivino, Conceptualization, Supervision, Funding acquisition, Writing - original draft, Project administration

### Author ORCIDs

Marianna Santopaolo ⓘ http://orcid.org/0009-0008-7593-287X
Michaela Gregorova ⓘ http://orcid.org/0000-0003-1605-0558
Elizabeth Oliver ⓘ http://orcid.org/0000-0002-1211-1942
Holly Baum ⓘ http://orcid.org/0000-0002-1311-6446
Andrew D Davidson ⓘ http://orcid.org/0000-0002-1136-4008
Linda Wooldridge ⓘ http://orcid.org/0000-0002-6213-347X
Laura Rivino ⓘ http://orcid.org/0000-0001-6213-9794

### Ethics

Human subjects: Information regarding our ethics approval and consent process is provided in the Materials and Methods section and copied below.Patients hospitalized with COVID-19 (≥18 years of age) were recruited between 30th March and 3rd June 2020 into the observational study DIagnostic and Severity markers of COVID-19 to Enable Rapid triage (DISCOVER), a single-centre prospective study based in Bristol (UK). Research Ethics Committee (REC) approval: REC:20/YH/1021. Survivors were invited at 3, 8 and 12 months post admission to attend outpatient follow up clinics for a systematic clinical assessment (Arnold et al 2020). For those patients attending a face-to-face follow-up, consent was taken to collect samples for research purposes (blood for PBMC isolation, plasma and serum). When available serum collected from patients at admission was made available to the research team.

### Decision letter and Author response

Decision letter https://doi.org/10.7554/eLife.85009.sa1
Author response https://doi.org/10.7554/eLife.85009.sa2

## Additional files

### Supplementary files
• MDAR checklist

## Data availability

All data generated or analysed during this study are included in the manuscript files or supplementary files. Raw file (FCS files) for all flow cytometry data have been deposited in the FlowRepository, the link for access to the data is provided in the Material and Methods, Flow cytometry data analysis section. The code script and data for the analysis in Figure 6 are publicly available on GitHub (copy archived at *Hamilton, 2022*).

The following datasets were generated:

| Author(s) | Year | Dataset title | Dataset URL | Database and Identifier |
|---|---|---|---|---|
| Gregorova R | 2023 | Intracellular cytokine staining | http://flowrepository. org/id/FR-FCM-Z6HZ | FlowRepository, FR-FCM-Z6HZ |
| Gregorova R | 2023 | IL15R experiment | http://flowrepository. org/id/FR-FCM-Z6HY | FlowRepository, FR-FCM-Z6HY |
| Gregorova R | 2023 | T NK cell phenotyping | http://flowrepository. org/id/FR-FCM-Z6GL | FlowRepository, FR-FCM-Z6GL |
| Gregorova R | 2023 | Innate cell phenotyping | http://flowrepository. org/id/FR-FCM-Z6H7 | FlowRepository, FR-FCM-Z6H7 |

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
