## [Editor Report]

This is an important paper that presents convincing evidence that certain cellular and molecular immune fingerprints at day 30 post-infection are associated with severe prior infection and may be risk factors for prolonged symptoms. The work will be of broad interest to clinicians, immunologists, and virologists.

---

## [Decision Letter]

**Decision letter after peer review:**

Thank you for submitting your article "Prolonged T-cell activation and long COVID symptoms independently associate with severe disease at 3 months in a UK cohort of hospitalized COVID-19 patients" for consideration by *eLife*. Your article has been reviewed by 3 peer reviewers, and the evaluation has been overseen by a Reviewing Editor and Tadatsugu Taniguchi as the Senior Editor. The reviewers have opted to remain anonymous.

Essential revisions:

1) Provide context from other diseases or other viral infection papers about the mechanistic significance of observed statistically significant differences in T cell activation and cytokine levels. As these assays are not routinely used in the clinic, it is important to provide some context to demonstrate that observed differences are not just statistically significant, but also biologically significant.

2) Attempt to reanalyze the data using more specific and granular PASC outcome measures.

3) Consider the inclusion of more control samples (either from pre-infection or later timepoints or from uninfected controls) if available.

4) Consider the analysis of T cell phenotypes associated with anti-inflammatory and regulatory activity as these could plausibly be associated with protection against long COVID symptoms.

*Reviewer #3 (Recommendations for the authors):*

– Please mention in the intro that vaccines are protective against PASC even with breakthrough infection.

– Please acknowledge the lack of racial diversity as a limitation of the study.

– Please speculate on whether certain results may differ in vaccinated and then infected people.

– In sup Figure 1, it would be better to compare mild, mode, and severe at comparable timepoints to see if the resolution is complete.

– In Figure 1E, 2E, and 5F, comparisons should be between groups, not between subsets within a group.

– In Figure 1N & 2N, it would be preferable to use the same scale on the y-axes to highlight that some populations are rare despite being significantly different.

– "The described elevation in IL-18 in severe patients is not evident in the data": this statement does not seem congruent with the data shown in the figures.

– How do we know that persistent bystander activation is not driven in tissue and cells and then traffic to blood? Please elaborate on this.

---

## [Author Response]

Essential revisions:1) Provide context from other diseases or other viral infection papers about the mechanistic significance of observed statistically significant differences in T cell activation and cytokine levels. As these assays are not routinely used in the clinic, it is important to provide some context to demonstrate that observed differences are not just statistically significant, but also biologically significant.

Research on other viral infections and diseases has shown that changes in T cell activation and cytokine levels can be biologically significant and have potential mechanistic significance for disease pathogenesis and response to treatment. The role of T cell activation is likely to differ in different diseases and at different times of disease and it is important to study these events to gain better understanding of the role of T cell activation during viral infection.

In acute influenza infection persistent T cell activation was associated with patient death while T cell activation early during infection associated with good clinical outcomes (Wang et al. Nature Comms 2018).

In chronic infection, for example in HIV infection, increased levels of Ki-67, CD38, HLA-DR and Granzyme B have been associated with disease progression and mortality. These markers are also frequently used in clinical trials to assess the efficacy of novel therapies targeting T cell activation and function (Hazenberg et. AIDS. 2003; Giorgi et al. Journal of Acquired Immune Deficiency Syndromes 2002).

In patients with chronic Hepatitis C virus (HCV) infection, CXCR3-associated chemokines including CXCL10 (IP-10) are associated with disease outcomes and response to anti-viral therapy. Increased serum CXCL10 levels at baseline are prognostic of failure to respond to anti-viral therapy in chronic HCV patients. In addition, elevated levels of IL-18 and CXCL20 (MIP1a) after treatment are indicative of resistance to anti-viral therapy (reviewed in Fahey et al. Cellular and Molecular Immunology 2013).

Elevated levels of cytokines such as IL-4, IL-7, and IL-17A have been implicated in the pathogenesis of autoimmune diseases such as multiple sclerosis and rheumatoid arthritis. In these diseases, targeting cytokine signalling pathways has been an effective therapeutic approach (McInnes et al. The New England Journal of Medicine 2011).

2) Attempt to reanalyze the data using more specific and granular PASC outcome measures.

Thank you for this suggestion. In addition to the number of long COVID symptoms at 3 months, we have now included Short Form Survey 36 (SF-36) physical and mental component summary scores as additional outcome measures (see updated Figure 6- source data 1). We did not find significant associations between PCS or MCS scores and T-cell profiles or cytokines/chemokines in either unadjusted or adjusted analyses after FDR correction.

3) Consider the inclusion of more control samples (either from pre-infection or later timepoints or from uninfected controls) if available.

Thank you for this suggestion. We have now analysed the matched samples from the same patients at 12 months post admission (Figure 2—figure supplement 1). Our results show that expression of T cell activation markers (CD38, HLA-DR, granzyme B) and of CXCR3 (peripheral tissue homing marker) is significantly downregulated at 12 months compared to the 3 months timepoint, suggesting a recovery of T cells at this later timepoint. Unsupervised analysis using UMAP shows that T cell populations were very similar at 12 months between patients who experienced mild, moderate or severe disease. In contrast, UMAP analyses of T cell populations from the same patients at 3 months showed significant differences between patients who had recovered from severe compared to mild and moderate disease. This data also suggests that differences in T cell activation/cytokines observed at 3 months between patients with severe versus non severe disease is not driven by differences already present in these patients at “baseline” (pre COVID-19), as at 12 months the T cell profiles did not differ across the patient groups (see added paragraph in discussion, page 19).

4) Consider the analysis of T cell phenotypes associated with anti-inflammatory and regulatory activity as these could plausibly be associated with protection against long COVID symptoms.

Unfortunately, due to the limiting number of cells we had available we were unable to also study T cell phenotypes associated with anti-inflammatory and regulatory activity. We have acknowledged this as a limitation of the study (discussion, page 19).

Reviewer #3 (Recommendations for the authors):– Please mention in the intro that vaccines are protective against PASC even with breakthrough infection.

We have mentioned this and cited references to support this (page 4).

– Please acknowledge the lack of racial diversity as a limitation of the study.

This has now been acknowledged (page 19)

– Please speculate on whether certain results may differ in vaccinated and then infected people.

All our patients were unvaccinated at the 3 months time point when long COVID symptoms were assessed hence we have not discusse3d potential differences between vaccinated and non vaccinated individuals.

– In sup Figure 1, it would be better to compare mild, mode, and severe at comparable timepoints to see if the resolution is complete.

Thanks for this comment. Our figure is designed to show differences in time within each severity group, to analyse resolution in time. Hence, we feel that the figure is more informative if data are shown from acute infection to resolution within each severity group rather than comparing mild, moderate and severe patients on the same graph.

– In Figure 1E, 2E, and 5F, comparisons should be between groups, not between subsets within a group.

Thanks for your comment – we have made comparisons between groups (using One-way ANOVA Kruskal-Wallis test with Dunn’s correction for multiple testing) and there were no statistical significances between the groups. Hence, we feel the data is more informative as currently displayed.

– In Figure 1N & 2N, it would be preferable to use the same scale on the y-axes to highlight that some populations are rare despite being significantly different.

Thanks for your comment, here (Figures1N, 2L) we want to show how each FlowSOM population differs between patients with mild, moderate and severe disease. Since the frequencies for each population are very different, we need to use a different axis for each population. If we use the same axis for each population we will not be able to show these differences.

– "The described elevation in IL-18 in severe patients is not evident in the data": this statement does not seem congruent with the data shown in the figures.

Based on Figure 3—figure supplement 1 we see minimal elevation of IL-18 in severe compared to mild and moderate patients (which was not statistically significant). Therefore, we feel that we cannot make this claim.

– How do we know that persistent bystander activation is not driven in tissue and cells and then traffic to blood? Please elaborate on this.

We agree that bystander activation could indeed occur within tissues. In line with this, activated CD4^+^ and CD8^+^ T cells at 3 months express the tissue homing receptor CXCR3. Our new analysis of T cells at 12 months shows that expression of CXCR3 as of activation markers is now undetectable at 12 months in both CD4^+^ and CD8^+^ T cells.